# Multi-antigen intranasal vaccine protects against challenge with sarbecoviruses and prevents transmission in hamsters

Ankita Leekha[1,7], Arash Saeedi[1,7], K M Samiur Rahman Sefat[1], Monish Kumar[1], Melisa Martinez-Paniagua[1], Adrian Damian[1], Rohan Kulkarni[1], Kate Reichel[1], Ali Rezvan[1], Shalaleh Masoumi[2], Xinli Liu[2], Laurence J. N. Cooper[3], Manu Sebastian[3], Courtney M. Sands[4], Vallabh E. Das[5], Nimesh B. Patel[5], Brett Hurst ®[6] & Navin Varadarajan ®[1] ✉

Immunization programs against SARS-CoV-2 with commercial intramuscular vaccines prevent disease but are less efficient in preventing infections. Mucosal vaccines can provide improved protection against transmission, ideally for different variants of concern (VOCs) and related sarbecoviruses. Here, we report a multi-antigen, intranasal vaccine, NanoSTING-SN (NanoSTING-Spike-Nucleocapsid), eliminates virus replication in both the lungs and the nostrils upon challenge with the pathogenic SARS-CoV-2 Delta VOC. We further demonstrate that NanoSTING-SN prevents transmission of the SARS-CoV-2 Omicron VOC (BA.5) to vaccine-naïve hamsters. To evaluate protection against other sarbecoviruses, we immunized mice with NanoSTING-SN. We showed that immunization affords protection against SARS-CoV, leading to protection from weight loss and 100% survival in mice. In non-human primates, animals immunized with NanoSTING-SN show durable serum IgG responses (6 months) and nasal wash IgA responses cross-reactive to SARS-CoV-2 (XBB1.5), SARS-CoV and MERS-CoV antigens. These observations have two implications: (1) mucosal multi-antigen vaccines present a pathway to reducing transmission of respiratory viruses, and (2) eliciting immunity against multiple antigens can be advantageous in engineering pan-sarbecovirus vaccines.

Humanity has undertaken one of the largest vaccination campaigns to protect all people against the respiratory virus SARS-CoV-2 and coronavirus-induced disease (COVID-19). mRNA (e.g., BNT162b2 and mRNA-1273) and adenovirus vector vaccines (e.g., ChAdOx1 nCoV-19) have been delivered intramuscularly (IM) to billions of recipients[1,2]. The evolution of variants of concern (VOC) like the Omicron VOCs has caused increase in infections even in countries with high vaccination coverage[3]. This increased frequency of infections combined with laboratory data that supports increased infectivity and immune escape by the variants has seeded concerns that we will end up in the cumbersome perpetual cycle of immunization trying to keep pace with evolving variants[4–7].

There are two primary concerns with the commercial vaccines against SARS-CoV-2. First, while the IM route of administration elicits

[1]William A. Brookshire Department of Chemical and Biomolecular Engineering, University of Houston, Houston, TX, USA. [2]Department of Pharmacological and Pharmaceutical Sciences, College of Pharmacy, University of Houston, Houston, TX, USA. [3]AuraVax Therapeutics, Houston, TX, USA. [4]Animal Care Operations, University of Houston, Houston, TX, USA. [5]College of Optometry, University of Houston, Houston, TX, USA. [6]Institute of Antiviral Research, Utah State University, UT Logan, USA. [7]These authors contributed equally: Ankita Leekha, Arash Saeedi. ✉e-mail: nvaradar@central.uh.edu

robust systemic immunity leading to the prevention of disease, they do not prevent viral infection in the upper airways. Unsurprisingly, the IM vaccines have demonstrated variable protection against upper-airway infection in preclinical models, with some offering no protection[8,9]. In humans, this led to both vaccinated and unvaccinated people harboring virus in the nostrils that facilitates transmission even by immunized individuals[10,11]. Moreover, the ability of the upper airways to serve as reservoirs facilitates viral evolution, and with the waning of vaccine-induced immunity over time, can enable the priming of new infections in vaccinated hosts[12,13]. The second concern is that the Spike (S) protein dominates the vaccine landscape against SARS-CoV-2 as the immunogen[14]. Since the S protein is essential for viral entry into host cells, it serves as the preferred target for eliciting neutralizing antibodies[14,15]. Although correlates of vaccine-induced protection have not been established, there is strong evidence that neutralizing antibodies, and specific antibodies targeting the receptor-binding domain (RBD) of the S protein, are likely predictors of vaccine efficacy and disease prevention[16,17]. The S protein, however, by being on the surface of the virion, is under constant evolutionary pressure to escape the host immune system while preserving viral entry[18,19]. Unsurprisingly, as the virus has spread globally, variants less susceptible to antibodies elicited by vaccines have evolved, necessitating modified vaccine manufacturing and continued booster immunizations[19–21].

Among other potential viral protein targets, the nucleocapsid (N) protein is expressed at elevated levels during infection and is highly immunogenic[22–24]. Studies tracking convalescent patient sera confirm robust antibody and T-cell responses against the N protein[25–28]. The primary function of the N protein is to package the viral genome into ribonucleoprotein complexes and to facilitate transcription while promoting escape from innate immunity (suppression of type I interferons, IFNs)[22]. Since the N protein performs multiple essential functions for the virus, it tends to accumulate fewer mutations resulting in the N protein of SARS-CoV-2 having 90% homology to SARS-CoV[29]. These attributes make the N protein a candidate for vaccine-induced immunity[23,24,30]. Indeed, T-cell-dependent mechanisms can confer at least partial protection against the original Wuhan strain after IM vaccine candidates immunizing with the N protein[31]. However, preclinical studies have shown that transfer of the anti-N immune sera failed to protect against SARS-CoV-2 infection in an adapted mouse model[32] which is consistent with antibodies against the N protein not being neutralizing as this protein is unassociated with viral entry. Furthermore, intradermal vaccination with the SARS-CoV N protein worsened infection and pneumonia due to T helper 2 (Th2) cell-biased responses[33]. This concern of enhanced respiratory disease mediated by Th2 responses has shifted the focus away from the SARS-CoV-2 N protein-based vaccines despite the potential for protective T-cell responses.

Mucosal vaccines can stimulate robust systemic and mucosal immunity, but the quality and quantity of the immune response elicited upon mucosal vaccination depend on the appropriate adjuvant. We had previously reported that liposomally encapsulated endogenous STING (stimulator of interferon genes) agonist (STINGa, 2′–3′ cGAMP), termed NanoSTING, functions as an excellent mucosal adjuvant that elicits strong humoral and cellular immune responses upon intranasal vaccination[34]. Here, we report that a multi-antigen intranasal subunit vaccine, NanoSTING-SN, delivers multi-factorial immunity by eliminating the virus from the nose and lung and prevents transmission to naïve animals. Our data provide a pathway to eliminating transmission of highly infectious variants and for engineering the next-generation vaccines that can protect against sarbecoviruses.

## Results

### Preparation and characterization of NanoSTING-S vaccine

NanoSTING is a liposomal adjuvant that comprises pulmonary surfactant-biomimetic nanoparticle formulated STINGa and enables mucosal immunity (Supplementary Fig. 1A)[34,35]. We synthesized NanoSTING, and dynamic light scattering (DLS) showed that the mean particle diameter of NanoSTING was 137 nm, with a polydispersity index (PDI) of 24.5% (Supplementary Fig. 1B) and a mean zeta potential of −63.5 mV (Supplementary Fig. 1C). We confirmed the ability of NanoSTING to induce IFN responses (IRF) using the THP-1 monocytic cells modified to conditionally secrete luciferase downstream of an IRF promoter. We stimulated THP-1 dual cells with NanoSTING and measured luciferase activity in the conditioned supernatant (Supplementary Fig. 1D) and showed that secretion was maximal at 24 h. We used recombinant trimeric S-protein to formulate the vaccine based on the SARS-CoV-2 B.1.351 (Beta VOC) as the immunogen[36] (Fig. 1A). SDS-PAGE under reducing conditions showed that the protein migrated between 180 and 250 kDa, confirming extensive glycosylation (Supplementary Fig. 2A). Upon incubation with NanoSTING, the S protein was adsorbed onto the liposomes with NanoSTING-S (NanoSTING-Spike) displaying a mean particle diameter of 144 nm (PDI 25.9%), and a mean zeta potential of −54.8 mV (Supplementary Fig. 2B, C). Unlike the trimeric S protein known to aggregate in solution, we tested NanoSTING-S after 9 months of storage at 4 °C. We found no evidence of aggregation, concluding that the vaccine formulation is stable at 4 °C (Supplementary Fig. 2D, E).

### Single-dose immunization of mice with NanoSTING-S vaccine yields cross-reactive humoral and cellular immunity against SARS-CoV-2

We immunized *BALB/c* mice with a single intranasal dose of NanoSTING-S (Fig. 1B) and observed no clinical symptoms, including weight loss, during the entire 28-day period (Supplementary Fig. 3). We conducted ELISA on day 28 to quantify binding to both full-length S proteins and the RBDs, with the latter serving as a surrogate for neutralization[37–39]. We observed robust serum IgG titers not only against Beta (B.1.351) but also against Alpha (B.1.1.7), Gamma (P.1), Delta (B.1.617.2), and Omicron (B.1.1.529) S proteins. We also observed high serum IgG titers against the RBDs from both the Beta and Alpha VOCs and high IgG titers against the full-length Beta and Delta spike proteins in bronchoalveolar lavage fluid (BALF, Fig. 1C, D). We evaluated the durability of the response upon vaccination at 5 months after immunization and confirmed high serum IgG titers (Supplementary Fig. 4). As IgA-mediated protection is an essential component of mucosal immunity for respiratory pathogens, we confirmed the role of intranasal NanoSTING-S as a mucosal vaccine candidate. We detected elevated serum IgA responses against all spike protein variants tested, although BALF IgA titers against full-length delta spike protein were weaker (Fig. 1E, F). At day 28, immunized mice showed robust and significant Th1/Tc1 responses by ELISPOT in both the spleen and the lung (Fig. 1G, H). We stimulated the spleen and lung cells with a pool of peptides containing mutations (B.1.351) in the S protein that differs from the Wuhan S protein. We observed a significant Th1 response against these mutation-specific S peptides, confirming a broad T-cell response that targets both the conserved regions and the mutated regions of the S protein (Fig. 1G, H). In contrast to the Th1/Tc1 responses, the Th2 responses were weaker but detectable (Fig. 1G, H). Collectively, these results established that NanoSTING acts as a mucosal adjuvant and that even a single-dose immunization with NanoSTING-S yielded robust IgG, IgA, and Th1/Tc1 responses that are cross-reactive against multiple VOCs.

### NanoSTING-S elicited immune responses confer protection against the Delta VOC

The Syrian golden hamster (*Mesocricetus auratus*) challenge model was used to assess the protective efficacy of NanoSTING-S. This animal model replicates COVID-19 severe disease in humans with infected animals demonstrating rapid weight loss, very high viral loads in the lungs, extensive lung pathology, and even features of long COVID[40,41].

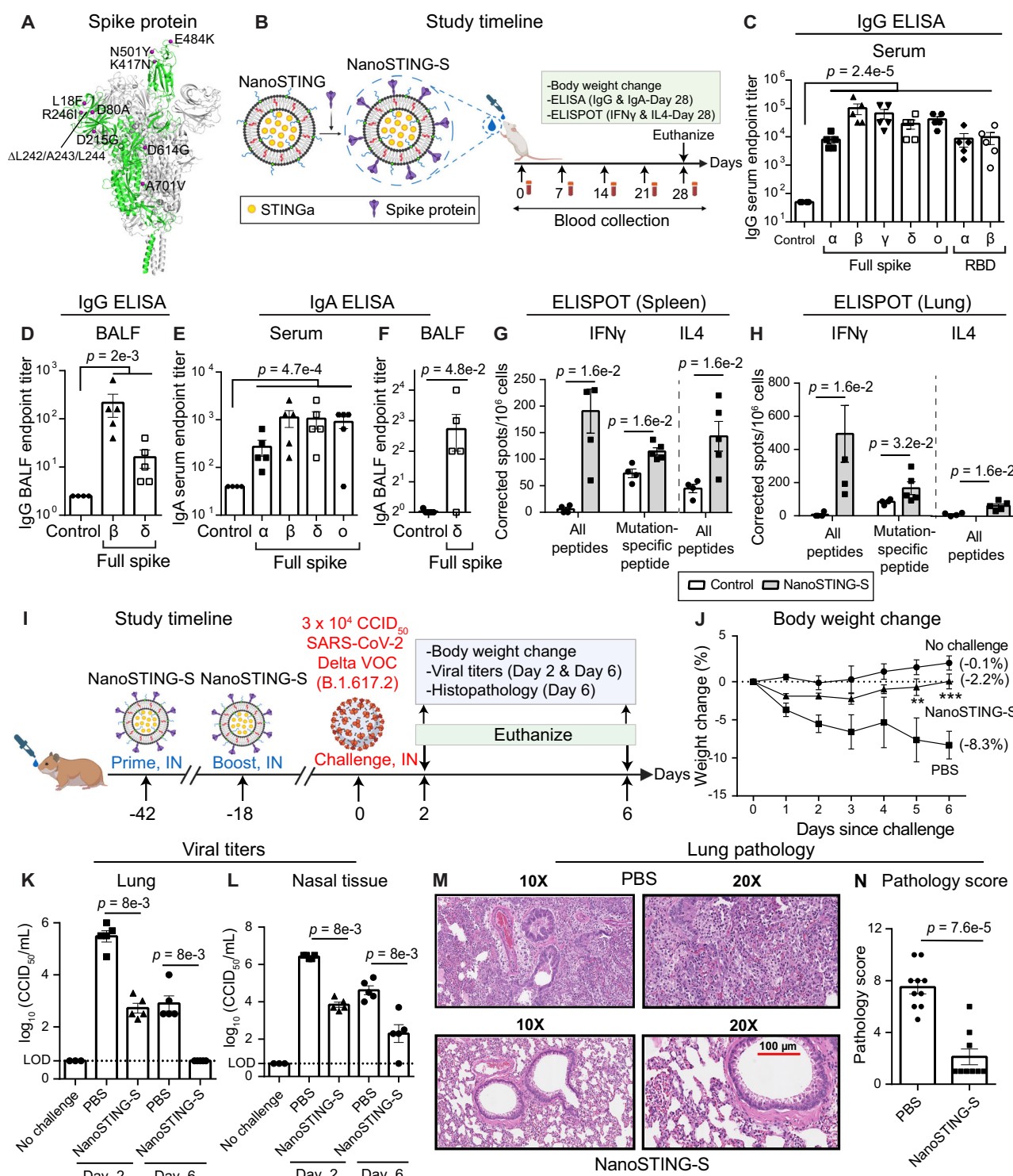

**Figure (panels A–N)**

Additionally, unlike the K18-hACE2 transgenic mouse model, hamsters recover from the disease and hence offer the opportunity to study the impact of treatments in the lungs (disease) and nasal passage (transmission)[40,42]. We chose the SARS-CoV-2 Delta VOC to infect the animals for two reasons: (1) this VOC causes severe lung damage, and (2) Delta-specific S-mutations, including L452R and T478K within the RBD, are absent in our immunogen (Fig. 1A), providing an opportunity to assess cross-protection. We administered two doses of intranasal NanoSTING-S 24 days apart to hamsters, which were subsequently challenged with the Delta VOC through the intranasal route (Fig. 1I). Animals in the sham-vaccinated group showed a mean peak weight

loss of $8 \pm 2\%$. By contrast, animals vaccinated with NanoSTING-S were largely protected from weight loss (Fig. 1J), mean peak weight loss of $2.3 \pm 0.7\%$, consistent with the results obtained by adenovirally vectored IM vaccines challenged with either the Wuhan or Beta strains[43]. We sacrificed half of the animals on day 2 (peak of viral replication) and the other half on day 6 (peak of weight loss in unimmunized animals) to quantify viral titers. NanoSTING-S reduced infectious viral loads in the lung by 300-fold by day 2 compared to sham-vaccinated animals, and by day 6, the infectious virus was undetectable in all animals (Fig. 1K). Viral replication in the lung of the animal models clinical human disease and death, while viral replication in the nasal

**Fig. 1 | NanoSTING-S vaccine yields cross-reactive humoral and cellular immunity in mice and provides protective efficacy against Delta VOC in hamsters. A** 3D structure of trimeric S protein (B.1.351) with the twelve mutations indicated (PDB: 7VX1)[75]. **B** Study timeline: We immunized *BALB/c* mice (*n* = 5/group) with a single dose of NanoSTING-S intranasally, followed by the collection of serum every week. We monitored the body weights of the animals every week after the immunization. We euthanized the animals by cervical dislocation at day 28 and then collected BALF, serum, lungs, and spleen. Primary endpoints were the body weight change, ELISA (IgG & IgA), and ELISPOT (IFNγ and IL4). Naïve *BALB/c* mice were used as controls (*n* = 4/group). **C–F** Humoral immune responses in the serum and BALF were evaluated using S-protein-based IgG & IgA ELISA. Splenocytes (**G**) or lung cells (**H**) were stimulated ex vivo with overlapping peptide pools, and IFNγ & IL4 responses were detected using an ELISPOT assay. **I** Experimental setup for challenge study in hamsters: We immunized Syrian golden hamsters (*n* = 10/group) intranasally with two doses of NanoSTING-S (first dose at day -42 and second dose at day -18, and challenged the hamsters intranasally with $3 \times 10^4$ CCID$_{50}$ of the SARS-CoV-2 Delta VOC on day 0. Post challenge, we monitored the animals for 6 days for changes in their body weight. We euthanized half of the hamsters on day 2 and the other half at day 6 for histopathology of the lungs, with viral titers of lung and nasal tissues measured on day 2 and day 6. **J** Percent body weight change of hamsters compared to the baseline at the indicated time intervals. **K, L** Viral titers were measured by end-point titration assay in lungs and nasal tissues post-day 2 and day 6 of infection. The dotted line indicates the limit of detection of the assay (LOD).

**M, N** Pathology score and a representative hematoxylin and eosin (H & E) image of the lung showing histopathological changes in hamsters treated with NanoSTING-S and PBS; all images were acquired at 10× & 20×; scale bar, 100 μm. Individual data points represent independent biological replicates taken from separate animals; vertical bars show mean values with error bars representing SEM. Each dot represents an individual mouse/hamster. For (**C–H, K, L, N**), the analysis was performed using two-tailed Mann-Whitney U-test: ****$p < 0.0001$; ***$p < 0.001$; **$p < 0.01$; *$p < 0.05$; ns: not significant. For (**J**), the data was compared via mixed-effects model for repeated measures analysis. Lines depict group mean body weight change from day 0; error bars represent SEM. For (**J**), the exact *p* values comparing the NanoSTING-S group to the PBS group are Day 5: $p = 5e-3$, Day 6: $p = 3.5e-4$. Asterisks indicate significance compared to the PBS-treated animals at each time point. Data presented as combined results from two independent experiments [**A–H**: Immunogenicity study with NanoSTING-S, **I–N**: Challenge study with Delta VOC], each involving one independent animal experiment. Gender was not tested as a variable, and only female mice were included in the study (**A–H**). Gender was tested as a variable with equal number of male and female hamsters included in study (**I–N**). See also Supplementary Figs. 1, 2, 3. **B, I** Created with BioRender.com released under a Creative Commons Attribution-NonCommercial-NoDerivs 4.0 International license (https://creativecommons.org/licenses/by-nc-nd/4.0/deed.en). Abbreviations - IN Intranasal. Number of animals used: **A–H**: *n* = 4–5/group, **I–N**: *n* = 10/group Source data are provided as a Source Data file.

compartment models human transmission. Immunization with NanoSTING-S reduced infectious viral loads in the nasal compartment by 380-fold by day 2 compared to unimmunized animals. By day 6, vaccinated animals showed a further significant reduction in the infectious virus (Fig. 1L). To examine the pathobiology of viral infection, we analyzed the lung tissue on day 6 after the challenge using an integrated scoring rubric (range from 1-12) to quantify host immune response and disease severity. We recorded immune cell infiltration and widespread viral pneumonia in the lungs of sham-vaccinated hamsters, whereas vaccinated animals revealed minimal evidence of invasion by inflammatory cells or alveolar damage (Fig. 1M, N). In aggregate, hamsters vaccinated with NanoSTING-S when challenged with the Delta VOC were protected in the lung against heterologous VOC and partially protected in the nasal passage. The reduction in viral loads in the nasal compartment suggests an advantage of mucosal vaccination to reduce transmission of the virus[44].

## Modeling of the immune response against both S- and N-proteins predicts synergistic protection

The results from the NanoSTING-S experiments demonstrated that the immune responses protect against disease in the lung but are insufficient to eliminate viral infection/replication in the nasal passage as a surrogate for transmission. To further bolster the protection against viruses, we explored additional antigens. We specifically chose the N protein because it is an abundantly expressed soluble and immunogenic protein. A mathematical model was used to help understand if a multi-antigen vaccine comprising both S- and N-proteins (NanoSTING-SN) can offer improved protection[45]. We explored the parameter space of an established model describing viral kinetics in the nasal passage obtained by fitting longitudinal viral titers from infected patients (Fig. 2A)[46]. The vaccine-induced neutralizing antibody responses against the S-protein serve as de-novo blockers of viral entry and impede viral production through immune effector mechanisms. We modeled a range (40–100%) of vaccine efficacies (directed only against the S-protein) to account for the differences in protection, specifically in the nasal compartment, and investigated the influence on viral elimination. The model revealed a reduction in viral load between 35% and 90% when the S-vaccine efficacy in the nasal compartment varied from 40 to 80% (Fig. 2B). An anti-S efficacy of >80% in the nasal compartment is difficult to accomplish even with mucosal immunization (some IM vaccines offer no significant nasal immunity). Hence, it can explain the inability of vaccines targeting only the S protein to

prevent nasal replication[8,9]. Next, we modeled a mucosal vaccine based exclusively on the N-protein. For the mucosal N-protein vaccine, we anchored to a mechanism of protection through the induction of cytotoxic T-cell responses that kill virally infected cells, thus reducing the number of cells capable of producing/propagating the virus. Under this scenario, the model predicted that the killing rate constant of cytotoxic T cells (CTL) would have to be 8.5 per day to achieve a 99% reduction in viral loads (Fig. 2C). This value is at least 10-fold higher than the predicted/measured killing capacity of CD8+ T cells in vivo; hence, it is not surprising that single-antigen N-based vaccines do not confer protection[47,48]. To quantify if multi-antigen vaccines can offer synergistic protection, we modeled combined protection by including S-directed vaccines that offer partial protection (primarily antibody-mediated) in the nasal compartment with the cytotoxic T-cell responses against the N protein. We tested a range of S-protein vaccine efficacies (40–100%) in the nasal compartment in combination with cytotoxic N responses (Fig. 2D). The model predicted that a physiologically relevant CTL killing rate of 0.4 and 0.6 days per day would lead to a 1000- and 10,000-fold reduction in peak viral load, respectively, when the efficacy of the spike vaccine was only 80% [Fig. 2D (red box) and Supplementary Fig. 5]. Indeed, studies in humans infected with COVID-19 have demonstrated a robust and long-lived CTL response in the nasal compartment and that CD8$^+$ T cells specific for the N protein can directly inhibit viral replication[49,50]. Collectively, these results from modeling predicted that combination vaccines targeting S and N proteins can mediate synergistic protection in eliminating viral replication in the nasal compartment.

## Immunization with NanoSTING-SN vaccine yields balanced humoral and cellular immunity and eliminates virus in the lung and nasal compartments upon challenge with the SARS-CoV-2 Delta VOC

We formulated vaccines containing both antigens to test the model that the immune response against both the S and N proteins can be synergistic (Fig. 3A). We initially performed immunogenicity experiments in mice with 10 μg of each recombinant N and S proteins adjuvanted with NanoSTING. We observed that while 100% of animals seroconverted and showed IgG responses against the S protein, seroconversion against the N protein was variable (40–80%) [not shown]. We accordingly modified the mass ratio of N:S protein (2:1) and adjuvanted it with NanoSTING to formulate NanoSTING-SN (Fig. 3A). The NanoSTING-SN displayed a mean particle diameter of 142 nm (PDI

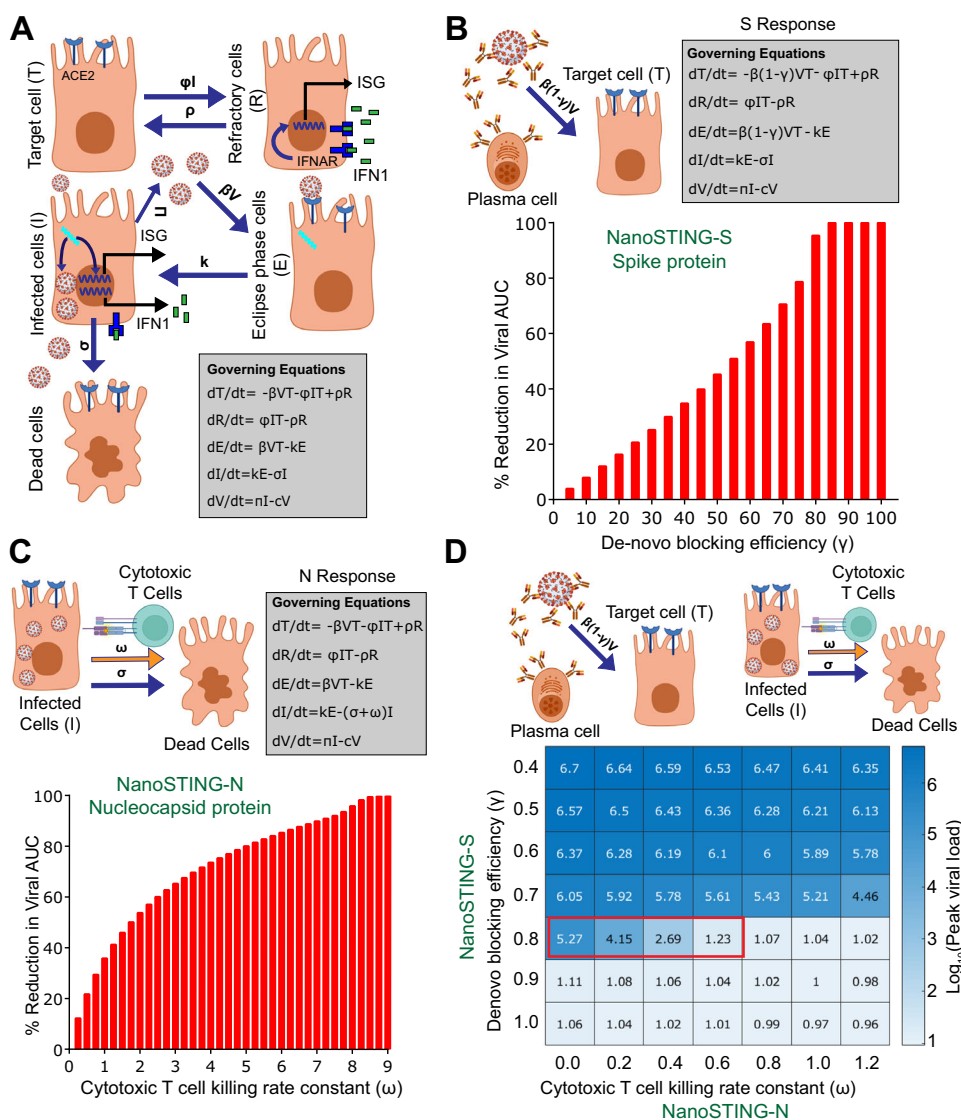

**Fig. 2 | Quantitative modeling of the combined immune response against both proteins predict synergistic protection. A** Schematic and governing equations describing viral dynamics without vaccination, with spike protein immunization, or nucleocapsid protein immunization (IFNAR interferon-α/β receptor, IFN1 type-I interferons, ISG interferon-stimulated gene). In the nasal compartment, SARS-CoV-2 (**V**) infects target epithelial cells (**T**) at the rate **βV**. The infected cells remain in an eclipse phase (**E**) before they become infected cells (**I**) with a rate constant (**k**) and start producing viral particles at rate **π**. The infected cells produce antiviral responses, which make the target cells refractory (**R**) with a rate constant directly proportional to the number of infected cells (**φI**). The infected cells die with a rate constant (**σ**). The refractory cells become target cells at rate (**ρ**). **B** Upon immunization with spike protein, the rate constant of target cell infection is reduced from **βV** to **βV**(1-γ) where γ is antibody-mediated blocking efficiency. The bar graph shows a percent reduction in viral area under the curve (AUC) with increasing de-novo blocking efficiency (antibodies against the spike protein). **C** Upon

immunization with N protein, the rate constant of elimination of infected cells is increased by **ω** due to the killing of infected cells by T cells. The bar graph shows a percent reduction in viral AUC upon cytotoxic T cell-mediated killing of infected cells. **D** Upon immunization with N and S protein the rate constant of elimination of infected cells is increased by **ω** and the rate constant of target cell infection is reduced from **βV** to **βV**(1-γ). The heatmap shows the effectiveness of the combined effect of de-novo blocking (S response) and T cell-mediated killing (N response). The red box indicates the synergistic effect of N and S response in achieving multifactorial immunity. See also Supplementary Fig. 5, Supplementary Methods, Sup Note 1. Parts of (**A**−**D**) were created with BioRender.com released under a Creative Commons Attribution-NonCommercial-NoDerivs 4.0 International license (https://creativecommons.org/licenses/by-nc-nd/4.0/deed.en). Abbreviations - ACE2 angiotensin-converting enzyme 2, ISG interferon-stimulated gene, IFN1 type-I interferons, IFNAR interferon-α/β receptor, AUC area under the curve. Source data are provided as a Source Data file.

26.2%) and a mean zeta potential of −48.4 mV (Supplementary Fig. 6A, B). We tested NanoSTING-SN after 9 months of storage at 4 °C, and confirmed that it displayed excellent stability, like the NanoSTING-S vaccine (Supplementary Fig. 6C, D). Single-dose intranasal vaccination in mice with NanoSTING-SN was safe (Supplementary Fig. 7) and yielded robust serum IgG titers against the N protein and full-length S protein variants at day 35 (Fig. 3B). We documented robust antigen-specific, cross-reactive IgG responses in the BALF (Fig. 3C) and observed cross-reactive IgA responses in the serum and BALF at day 35

(Fig. 3D, E). We measured Th1 and Th2 responses against the N- and S-proteins in both the spleen and the lung at day 51 (Fig. 3F, G) and observed no significant Th2 response (IL4) in both tissues. Based on these promising immunogenicity data in mice, we evaluated the protective efficacy of NanoSTING-SN in hamsters. We vaccinated hamsters intranasally with two doses of NanoSTING-SN and challenged the immunized hamsters with the Delta VOC through the intranasal route (Fig. 3H). Animals immunized with NanoSTING-SN were completely protected from weight loss (mean peak weight loss of 1 ± 1%) [Fig. 3I].

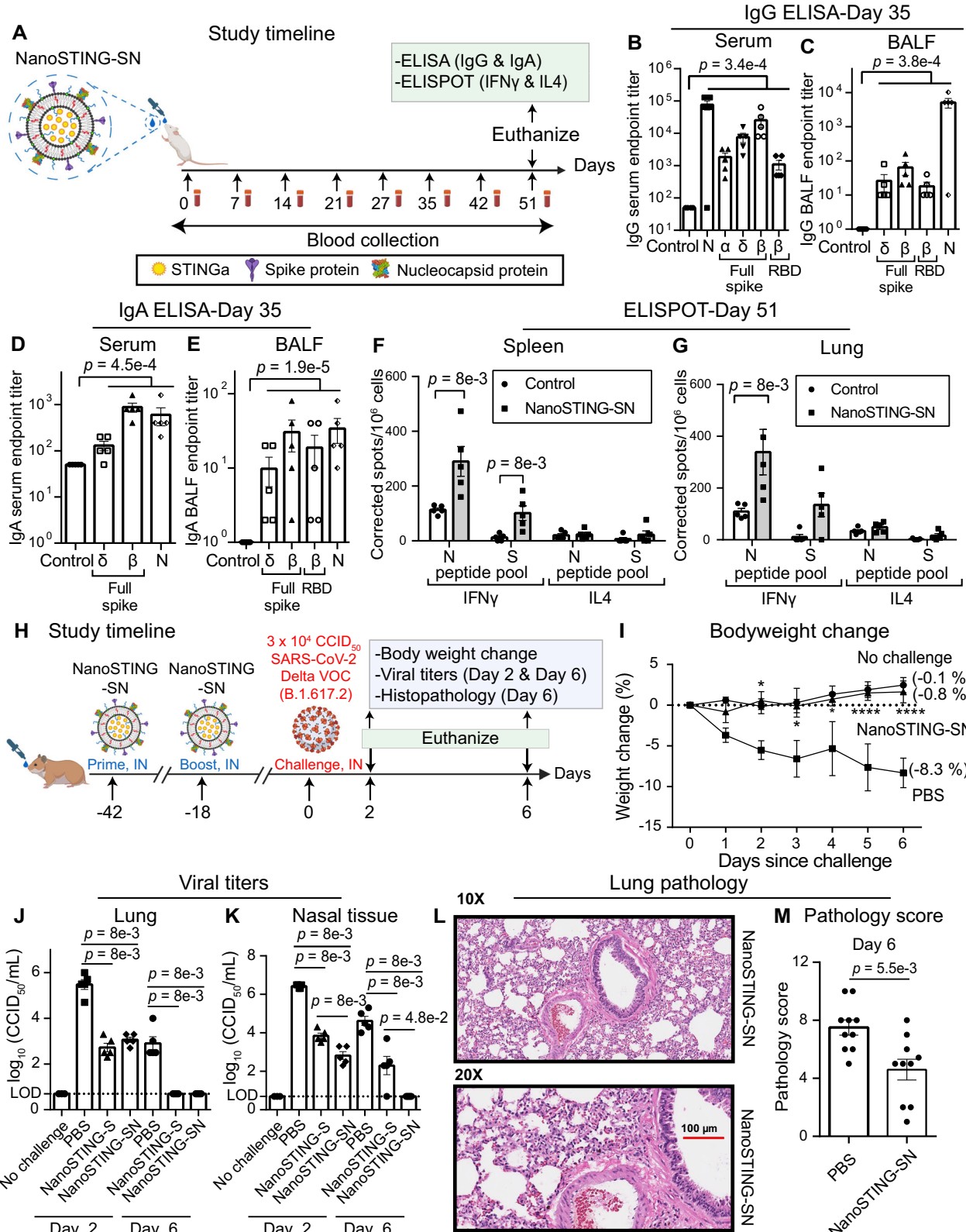

**A** Study timeline

NanoSTING-SN

-ELISA (IgG & IgA)
-ELISPOT (IFNγ & IL4)

Euthanize

Days

Blood collection

⬤ STINGa   🔱 Spike protein   🦠 Nucleocapsid protein

**B** IgG ELISA-Day 35 — Serum
*p* = 3.4e-4

**C** BALF
*p* = 3.8e-4

**D** IgA ELISA-Day 35 — Serum
*p* = 4.5e-4

**E** BALF
*p* = 1.9e-5

**F** ELISPOT-Day 51 — Spleen
*p* = 8e-3

**G** Lung
*p* = 8e-3

**H** Study timeline

NanoSTING-SN — Prime, IN
NanoSTING-SN — Boost, IN
3 x 10⁴ CCID₅₀ SARS-CoV-2 Delta VOC (B.1.617.2) — Challenge, IN

-Body weight change
-Viral titers (Day 2 & Day 6)
-Histopathology (Day 6)

Euthanize

**I** Bodyweight change

**J** Viral titers — Lung

**K** Nasal tissue

**L** Lung pathology

**M** Pathology score

Like the results of the NanoSTING-S vaccine, NanoSTING-SN eliminated viral replication in the lung by day 6 post-challenge (Fig. 3J), suggesting that S-specific immune responses are the dominant factor in providing immunity in the lung. In the nasal compartment, NanoSTING-SN showed a significant reduction in infectious viral particles by day 2, even in comparison to NanoSTING-S, and significantly, by day 6 there was a complete elimination of infectious viral particles

in the nasal tissue of the NanoSTING-SN vaccinated animals (Fig. 3K). Pathology also confirmed that vaccinated and challenged animals had minimal alveolar damage (Fig. 3L, M). Although the resolution of the inflammatory responses characterized by macrophages and lymphocytes was different when comparing NanoSTING-SN (multiple antigens) vs NanoSTING-S (single antigen) (Figs. 3M, 1N), viral titers in both the lung and nasal compartments were eliminated upon vaccination

**Fig. 3 | NanoSTING-SN vaccine yields balanced humoral and cellular immunity targeting both proteins and eliminates virus in both the lung and nasal compartments upon challenge with the SARS-CoV-2 Delta VOC. A** Experimental setup: We immunized two groups ($n = 5$/group) of mice by intranasal administration with NanoSTING-SN followed by serum collection every week. We monitored the body weights of the animals every week after the immunization until the end of the study. We euthanized the animals at day 51 followed by the collection of BALF, serum, lungs, and spleen. Body weight change, ELISA (IgG & IgA), and ELISPOT (IFNγ and IL4) were primary endpoints. Naïve *BALB/c* mice were used as controls ($n = 5$/group). **B–E** Humoral immune responses in the serum and BALF were evaluated using S-protein and N-protein based IgG & IgA ELISA. Splenocytes (**F**) or lung cells (**G**) were stimulated ex vivo with overlapping peptide pools, and IFNγ & IL4 responses were detected using an ELISPOT assay. **H** Timeline for challenge study done in Syrian golden hamsters: We immunized hamsters intranasally with two doses of NanoSTING-SN (first dose at day -42 and the second dose at day -18) and challenged the hamsters intranasally with $3 \times 10^4$ CCID$_{50}$ of the SARS-CoV-2 Delta VOC on day 0. Post-challenge, we monitored the animals for 6 days for changes in their body weight. We euthanized half of the hamsters on day 2 and the other half on day 6 for histopathology of the lungs, with viral titers of lung and nasal tissues measured on day 2 and day 6. **I** Percent body weight change of hamsters compared to the baseline at the indicated time intervals. **J, K** Viral titers were measured by end-point titration assay in lungs and nasal tissues post-day 2 and day 6 of infection. The dotted line indicates LOD. **L, M** A representative hematoxylin and eosin (H & E) image and pathology scores of the lung showing histopathological changes in hamsters treated with NanoSTING-SN and PBS; all images were acquired at 10x & 20×; scale bar, 100 μm. Individual data points represent independent biological replicates taken from separate animals; vertical bars show mean values with error bars representing SEM. Each dot represents an individual mouse/hamster. For (**B–G, J, K, M**), the analysis was performed using a two-tailed Mann-Whitney U-test: ****$p < 0.0001$; ***$p < 0.001$; **$p < 0.01$; *$p < 0.05$; ns: not significant. For (**I**), the data was compared via mixed-effects model for repeated measures analysis. Lines depict group mean body weight change from day 0; error bars represent SEM. For (**I**), the exact $p$ values comparing the NanoSTING-SN group to the PBS group are Day 2: $p = 1.9e\text{-}2$, Day 3: $p = 1.0e\text{-}2$, Day 4: $p = 2.0e\text{-}2$, Day 5: $p = 7.3e\text{-}5$, Day 6: $p = 1.3e\text{-}5$. Asterisks indicate significance compared to the PBS-treated animals at each time point. Data presented as combined results from two independent experiments [**A–G**: Immunogenicity study with NanoSTING-SN, **H–M**: Challenge study with Delta VOC], each involving one independent animal experiment. Gender was not tested as a variable, and only female mice were included in the study (**A–G**). Gender was tested as a variable with equal number of male and female hamsters included in the study (**H–M**). See also Supplementary Figs. 6, 7. **A, H** were created with BioRender.com released under a Creative Commons Attribution-NonCommercial-NoDerivs 4.0 International license (https://creativecommons.org/licenses/by-nc-nd/4.0/deed.en). Abbreviations: IN Intranasal. Number of animals used: **A–G**: $n = 4$–5/group, **H–M**: $n = 10$/group Source data are provided as a Source Data file.

with NanoSTING-SN. In aggregate, these results illustrate that NanoSTING-SN can provide complete elimination of the virus in both the lung and nasal compartments.

## Immunization with NanoSTING-N yields durable humoral and cellular immunity but is not sufficient to confer protection against Delta VOC

To quantify the role of anti-N immunity in mucosal protection, we characterized the immune response elicited against the N protein by formulating NanoSTING-N (NanoSTING-Nucleocapsid) and testing it in mice. Independent studies with K18-hACE2 mice immunized with viral vector-based N protein and challenged the early lineage variants (Wuhan and Alpha VOC) revealed mixed results with either partial or a complete lack of protection[24,31]. The predicted structure of the SARS-CoV-2 N protein comprises an RNA binding domain, a C-terminal dimerization domain, and three intrinsically disordered domains that promote phase separation with nucleic acids[51] (Supplementary Fig. 8A). To confirm the size of the N-protein, we performed SDS-PAGE, wherein we demonstrated that the recombinant N protein had an estimated molecular mass of 47 kD (Supplementary Fig. 8B). We confirmed the functional activity of the protein by assaying binding to plasmid DNA based on the quenching of the fluorescent DNA condensation probe DiYO-1 (Supplementary Fig. 8C)[52] with PEI (R) used as a positive control (Supplementary Fig. 8D). To formulate the vaccine, NanoSTING-N, we mixed the N protein with NanoSTING to allow the adsorption of the protein onto the liposomes (Fig. 4A). The formulated NanoSTING-N had a mean particle diameter of 107 nm (PDI 20.6%), and a mean zeta potential of −51 mV (Supplementary Fig. 8E, F). Although the recombinant N protein showed a strong propensity for aggregation upon storage at 4 °C for 6 months, NanoSTING-N was stable with no change in size or zeta potential (Supplementary Fig. 8G, H). Consistent with our NanoSTING-SN studies, we immunized two groups of mice by intranasal administration with either 10 μg (NanoSTING-N10) or 20 μg of N protein (NanoSTING-N20, Fig. 4A and Supplementary Fig. 9). The serum IgG responses at both doses were similar at day 27, although the IgG titers elicited by the NanoSTING-N20 were higher than NanoSTING-N10, the difference was not significant (Fig. 4B).

In contrast to vaccination with the trimeric NanoSTING-S (early response at day 7), the kinetics of IgG responses were delayed, and responses were only observed at day 14 (Supplementary Fig. 10). Both, NanoSTING-N10 and NanoSTING-N20 yielded antigen-specific IgG responses in the BALF and IgA response in serum (Fig. 4C, D). We examined the activation and function of N-protein-specific memory CD8$^+$ T cells in the lungs and spleen using granzyme B (GzB) and the activation-induced marker CD137. Restimulation ex vivo with a pool of overlapping peptides derived from the N protein resulted in a significant increase in the frequency of activated (CD8$^+$CD137$^+$), and cytotoxic (CD8$^+$GzB$^+$) T cells in the spleen (Fig. 4E, F, H, I) and to a lesser extent in the lung of both the NanoSTING-N10 and NanoSTING-N20 vaccinated mice (Supplementary Fig. 11B, C). The overall frequencies of the lung resident memory CD8$^+$CD103$^+$ and CD8$^+$CD103$^+$CD69$^+$ T cells were no different between the immunized animals and the control group (Supplementary Fig. 11D, E). NanoSTING-N10 and NanoSTING-N20 immunized mice showed robust and significant splenic and lung Th1/Tc1 responses (Fig. 4G, J). We did not observe a measurable IL4 (Th2) response upon immunization with NanoSTING-N10 and NanoSTING-N20 (Fig. 4G, J). To test the durability of the NanoSTING-N response, we immunized mice with NanoSTING-N20 and monitored the animals for 62 days (Supplementary Fig. 12A). NanoSTING-N20 vaccinated animals reported no weight loss (Supplementary Fig. 12B) and revealed robust serum IgG and IgA titers at day 62 (Supplementary Fig. 12C, D). We also confirmed that the N-reactive Th1 responses were conserved in the spleen and lung at day 62 (Supplementary Fig. 12E, F). These results can be Collectively, these results establish that immunization with NanoSTING-N results in IgG and IgA immune responses and long-lived Th1/Tc1 but not deleterious Th2 immune responses.

Based on the immunogenicity data in mice, we evaluated the protective efficacy of NanoSTING-N in hamsters. We intranasally vaccinated hamsters with two doses of NanoSTING-N and challenged the immunized hamsters with the SARS CoV-2 Delta VOC through the intranasal route (Fig. 4K). Animals in both the vaccinated and sham-vaccinated groups showed significant weight loss (Fig. 4L). Consistent with the lack of protection from weight loss, infectious viral titers were no different in the lung or nasal passage on either day 2 or day 6 in both vaccinated and sham-vaccinated animals (Fig. 4M, N). In addition, we observed that the aggregate pathology score of NanoSTING-N treated hamsters was not significantly different from sham-vaccinated animals, although the distribution of pathology scores appeared bimodal (Fig. 4O, P). Collectively, the immunization and the challenge data are aligned with our mathematical model and illustrate that while NanoSTING-N elicits strong Tc1 responses, these responses are insufficient to prevent viral expansion in the absence of S-directed immunity.

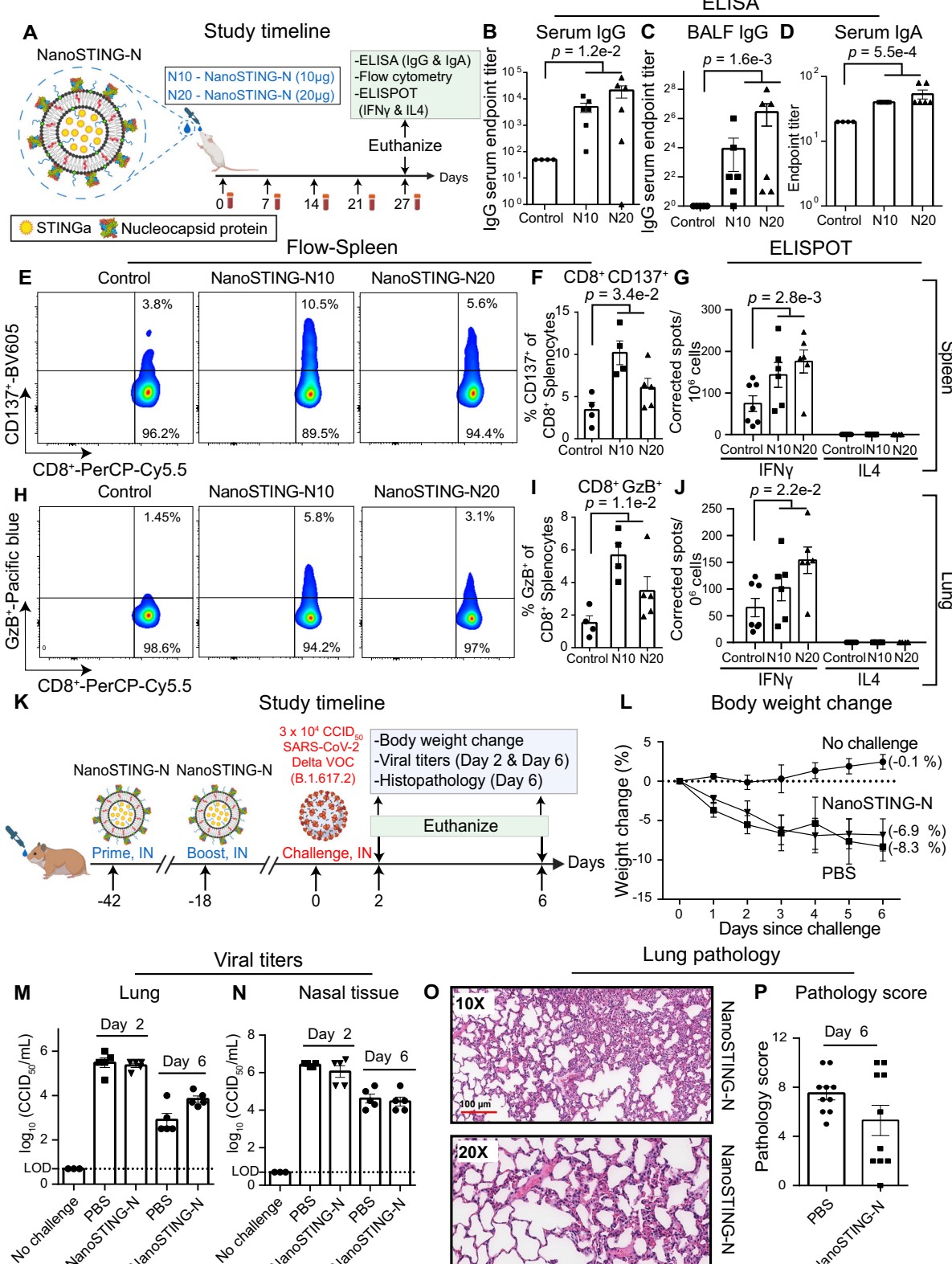

**Two doses of NanoSTING-SN abolishes transmission of SARS-CoV-2 Omicron VOC**

The SARS-CoV-2 Omicron VOCs are transmitted very efficiently, and we next wanted to directly investigate if immunization with NanoSTING-SN can prevent the transmission of this highly infectious VOC. We established a transmission experiment using the SARS-CoV-2 Omicron VOC (BA.5) and two groups of Syrian golden hamsters. For

group 1, the animals were sham immunized with PBS whereas for group 2, the animals were immunized with two doses NanoSTING-SN spaced by 21 days. These hamsters were challenged on day 35 with SARS-CoV-2 Omicron VOC (BA.5) and 1 day after the challenge; each animal was paired with a naïve unimmunized hamster (contact hamster) that allowed not only aerosol but also direct contact and fomite (including diet and bedding) transmission for 4 days (Fig. 5A).

**Fig. 4 | NanoSTING-N vaccine yields durable humoral and cellular immunity in mice but is insufficient to confer protection against the highly infectious Delta VOC in hamsters. A** Experimental setup: We immunized two groups (n = 5–6/group) of mice by intranasal administration with NanoSTING-N10 or NanoSTING-N20 followed by serum collection every week. We monitored the body weights of the animals every week after the immunization until the end of the study. We euthanized the animals at day 27 and then collected BALF, serum, lungs, and spleen. Body weight change, ELISA (IgG & IgA), flow cytometry (CD8+ T cells), and ELISPOT (IFNγ and IL4) were used as primary endpoints. Naïve *BALB/c* mice were used as controls (n = 4/group). **B, C** Humoral immune responses in the serum and BALF were evaluated using N-protein-based IgG ELISA. **D** Humoral immune responses in the serum were evaluated using N-protein-based IgA ELISA. Splenic CD8+ T cells were stimulated ex vivo with overlapping peptide pools, and (**E, F**) CD137 expression was quantified by flow cytometry (**G**) IFNγ & IL4 responses were detected using an ELISPOT assay. Splenic CD8+ T cells were stimulated ex vivo with overlapping peptide pools, and (**H, I**) GzB expression was quantified by flow cytometry (**J**) IFNγ & IL4 ESLIPOT from lung cells stimulated ex vivo with indicated peptide pools. **K** Experimental setup for challenge studies in Syrian golden hamsters. We immunized hamsters (n = 10/group) intranasally with two doses of NanoSTING-N (first dose at day -42 and the second dose at day -18 and challenged the hamsters intranasally with $3 \times 10^4$ CCID$_{50}$ of the SARS-CoV-2 Delta VOC on day 0. Post-challenge, we monitored the animals for 6 days for changes in body weight. We euthanized half of the hamsters on day 2 and the other half on day 6 for histopathology of the lungs, with viral titers of lung and nasal tissues measured on day 2 and day 6. **L** Percent body weight change of hamsters compared to the baseline at the indicated time intervals. **M, N** Viral titers were measured by end-point titration assay in lungs and nasal tissues post-day 2 and day 6 of infection. The dotted line indicates LOD. **O, P** Pathology score and a representative H & E image of the lung showing histopathological changes in hamsters treated with NanoSTING-N and PBS; all images were acquired at 10x & 20×; scale bar, 100 μm. Individual data points represent independent biological replicates taken from separate animals; vertical bars show mean values with error bars representing SEM. Each dot represents an individual mouse/hamster. For (**B–G, J, K, M**), the analysis was performed using two-tailed Mann-Whitney U-test: ****p < 0.0001; ***p < 0.001; **p < 0.01; *p < 0.05; ns: not significant. For (**I**), the data was compared via mixed-effects model for repeated measures analysis. Lines depict group mean body weight change from day 0; error bars represent SEM. Asterisks indicate significance compared to the PBS-treated animals at each time point. Data presented as combined results from two independent experiments [**A–J**: Immunogenicity study with NanoSTING-N, **K–P**: Challenge study with Delta VOC], each involving one independent animal experiment. Gender was not tested as a variable for the study, and only female mice were used (**A–G**). Gender was tested as a variable with an equal number of male and female hamsters included in the study (**H–M**). See also Supplementary Figs. 8–12. **A, K** were created with BioRender.com released under a Creative Commons Attribution-NonCommercial-NoDerivs 4.0 International license (https://creativecommons.org/licenses/by-nc-nd/4.0/deed.en). Abbreviations: GzB Granzyme, IN intranasal, N10 NanoSTING with 10 μg of Nucleocapsid protein, N20 NanoSTING with 10 μg of Nucleocapsid protein. Number of animals used: **A–J**: n = 4–6/group, **K–P**: n = 10/group Source data are provided as a Source Data file.

We quantified the viral titers in the index and contact hamsters in the lungs and nasal tissue. Vaccination protected the index hamsters from the virus, with only 2/8 hamsters showing a low amount of detectable virus in the lung (Fig. 5B) and nasal tissue (Fig. 5D). Vaccination completely blocked transmission in the lungs (Fig. 5C) and nasal tissue (Fig. 5E), and none of the contact hamsters showed detectable virus. These results suggest that at least early after immunization, two doses of NanoSTING-SN can prevent transmission of highly transmissible variants and long-term studies are warranted to quantify the duration of this protection.

We repeated these studies with a single dose of the vaccine and challenged with the SARS-CoV-2 Omicron VOC (BA.1.1.529). Even a single dose of the vaccine significantly reduced transmission to naïve animals (Fig. 6). These results demonstrated that even a single dose of NanoSTING-SN is effective at mitigating the transmission of the Omicron VOC, and two doses of the vaccine were sufficient to eliminate transmission, which has implications for controlling the outbreak of respiratory pathogens.

## NanoSTING-SN yields cross-reactive humoral immunity and confers protection against SARS-CoV

We next wanted to assess whether immunization with NanoSTING-SN would yield humoral responses that are cross-reactive against coronaviruses. Although the trimeric spike proteins from SARS-CoV, SARS-CoV-2, and MERS-CoV are structurally similar (Fig. 7A), sequence alignment of the RBDs of these same proteins showed significant sequence divergence (Fig. 7B). We evaluated the efficacy of NanoSTING-SN in mice by assaying the immune sera against both full-length S and N proteins and the RBDs from SARS-CoV-2, SARS-CoV, MERS-CoV (betacoronavirus) and HCOV-229E (alphacoronavirus) [Supplementary Fig. 13], with the RBD reactivity serving as a surrogate for neutralization. We documented robust antigen-specific, cross-reactive IgG responses against full-length S proteins and, as anticipated by the sequence divergence, a reduction in the reactivity to the RBDs of these different coronaviruses (Fig. 7C, D). By contrast, serum IgG ELISA against N proteins revealed similar titers against SARS-CoV-2 and SARS-CoV N proteins (Fig. 7C).

To test if the cross-reactive humoral activity against the S-RBD translated to protection against viral challenge, we tested the efficacy of dual dose intranasal NanoSTING-SN in mice against SARS-CoV (v2163 strain) [Fig. 7E]. Upon challenge with SARS-CoV, mice in the sham-vaccinated group showed a mean peak weight loss of 27 ± 1%. By contrast, animals vaccinated with NanoSTING-SN showed completely normal weights that were no different from unchallenged animals at all indicated time points (Fig. 7F). Survival data confirmed that NanoSTING-SN vaccinated mice showed 100% survival, illustrating a significant impact of treatment on survival of mice compared to the PBS group (Fig. 7G). These results demonstrated that NanoSTING-SN can provide complete protection against challenge from multiple sarbecoviruses.

Since our data with SARS-CoV-2 (Figs. 1J, 3I) illustrated that immune response against the spike protein elicited by NanoSTING-S was sufficient to protect the animals from weight loss, we investigated if the dominant protection afforded by NanoSTING-S would also translate to SARS-CoV. As expected, animals immunized with two doses of NanoSTING-S and NanoSTING-SN (Supplementary Fig. 14A) showed completely normal weights and 100% survival as compared to PBS-treated animals (Supplementary Fig. 14B, C).

## NanoSTING-SN confers durable humoral immunity in rhesus macaques

To assess the efficacy of NanoSTING-SN on Rhesus macaques (*M. mulatta*), we immunized three animals intranasally with two doses of NanoSTING-SN on Day 0 and Day 28 (booster dose). We monitored the animals for 44 days to track changes in body weight, attitude, appetite, and body temperature (Fig. 8A). None of the animals showed clinical signs such as loss of body weight (Fig. 8B) or increase in body temperature (Fig. 8C) upon administration of NanoSTING-SN. We evaluated humoral immunity induced in rhesus macaques using SARS-CoV-2 spike and nucleoprotein-specific IgG ELISA. We evaluated the efficacy of the dual dose of NanoSTING-SN by assaying the immune sera collected at day 21 and day 45 against both full-length S and N proteins and RBD from SARS-CoV-2 variants (BA.1, XBB1.5), SARS-CoV, and MERS-CoV. Consistent with our mice data, we documented robust antigen-specific, cross-reactive IgG responses against full-length S & N proteins from SARS-CoV-2 variants and from other coronaviruses at day 21 and post booster, the IgG titers were significantly increased at day 45 (Fig. 8D, E). Importantly, the IgG responses against both the S and N proteins did not reduce significantly over 6 months in the immunized NHPs (Non-human primates) (Supplementary Fig. 15).

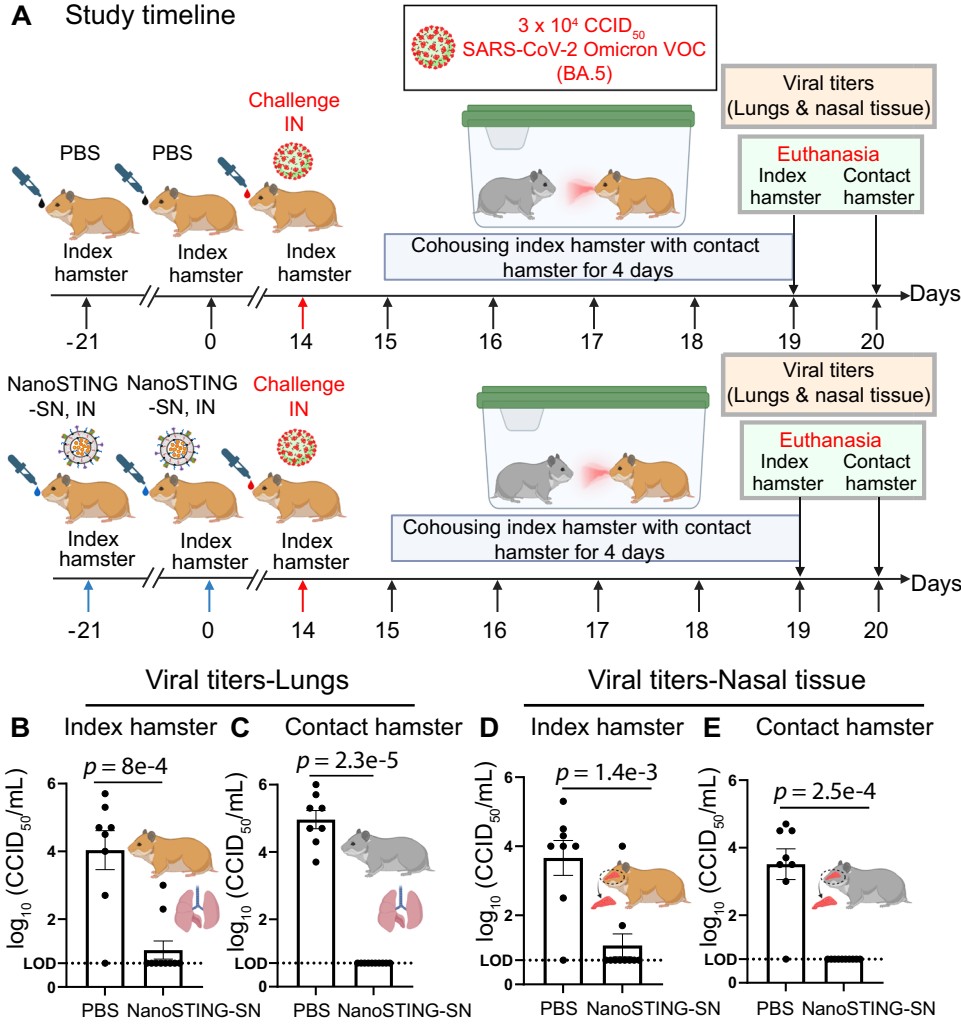

**Fig. 5 | Intranasal vaccination with NanoSTING-SN abolishes transmission of SARS-CoV-2 Omicron (BA.5) VOC in hamsters. A** Experimental setup: We immunized hamsters with a dual dose of the intranasal NanoSTING-SN vaccine ($n = 10$/group) or PBS ($n = 8$/group) 5 weeks (day-21) and 2 weeks (Day 0) prior to infection with $\sim3 \times 10^4$ CCID$_{50}$ of SARS-CoV-2 Omicron VOC (BA.5) [Day 14]. One day after the viral challenge, we co-housed the index hamsters in pairs with contact hamsters for 4 days in clean cages. We euthanized the index hamsters on day 4 of cohousing and contact hamsters on day 5 of cohousing. Viral titers in the lungs of the index and contact hamsters were used as primary endpoints. **B, C** Infectious viral particles in the lung tissue of contact and index hamsters at day 5 after viral administration post-infection were measured by end-point titration assay. The dotted line indicates LOD. **D, E** Infectious viral particles in the nasal tissue of contact and index hamsters at day 5 after viral administration post-infection were measured by end-point titration assay. The dotted line indicates LOD. Individual data points represent independent biological replicates taken from separate animals; vertical bars show mean values with error bars representing SEM. Each dot represents an individual hamster. The analysis was performed using two-tailed Mann-Whitney U-test: ****$p < 0.0001$; ***$p < 0.001$; **$p < 0.01$; *$p < 0.05$; ns not significant. Asterisks indicate significance compared to the PBS-treated animals at each time point. Data presented as combined results from one independent experiment. Gender was tested as a variable, and an equal number of male and female hamsters were included in the study. **A** and parts of (**B–E**) were created with BioRender.com released under a Creative Commons Attribution-NonCommercial-NoDerivs 4.0 International license (https://creativecommons.org/licenses/by-nc-nd/4.0/deed.en). Number of animals used: $n = 8$–10/group Source data are provided as a Source Data file.

To test mucosal immunity, we evaluated IgA responses after both the prime and the boost immunizations. We detected elevated cross-reactive serum IgA responses against full-length N, RBD, and all spike protein variants tested from SARS-CoV-2, SARS-CoV, and MERS (Fig. 8F, G). In humans, although serum IgG levels were associated with the prevention of disease against SARS-CoV-2 variants, newer data suggest that IgA antibodies in the nasal compartment correlate strongly with protection against infection, especially against the Omicron VOCs, and may serve as a surrogate for protection[53,54]. Accordingly, we evaluated the nasal wash IgA titers elicited upon immunization of the NHPs with NanoSTING-SN. We measured detectable and significant IgA responses against the S and N proteins of SARS-CoV-2 and the S protein of MERS-CoV (Fig. 8H). These results

illustrate that NanoSTING-SN elicits cross-reactive mucosal responses in the nasal compartment of rhesus macaques.

## Discussion

The continued evolution of SARS-CoV-2 and the potential for recombination between SARS-CoV-2 and closely related coronaviruses in bats has prompted an urgency in the development of vaccines targeting sarbecoviruses and coronaviruses[55,56]. Our study makes an essential contribution to the next-generation of pan-sarbecovirus and coronavirus vaccines. In this context, we report four significant results with NanoSTING-SN: (1) it yields both mucosal and systemic immunity against SARS-CoV-2 in hamsters against Delta VOC challenge, (2) it was sufficient to completely block transmission of the highly transmissible

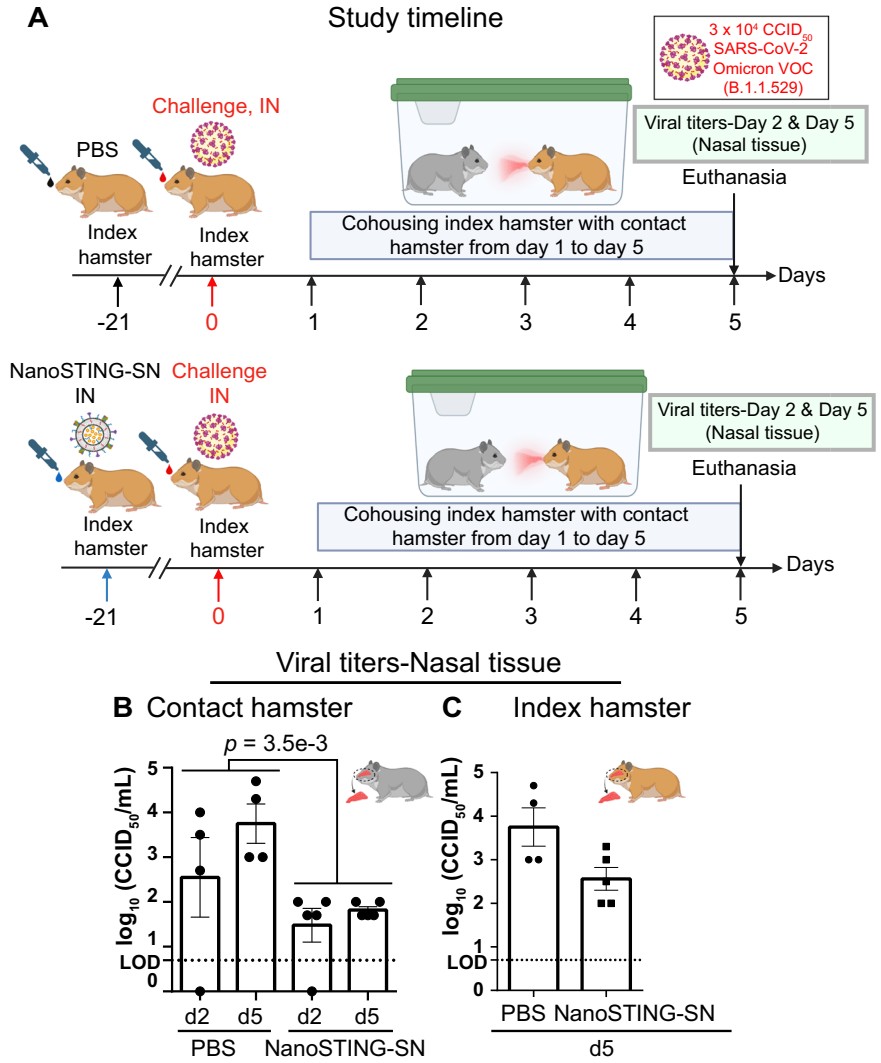

**Fig. 6 | Single dose intranasal administration of NanoSTING-SN limits transmission and viral replication of SARS-CoV-2 Omicron (B.1.1.529) VOC in hamsters. A** Experimental setup: We immunized hamsters with a single dose of the intranasal NanoSTING-SN ($n = 10$/group) vaccine or PBS ($n = 8$/group) 3 weeks prior to infection with ~$3 \times 10^4$ CCID$_{50}$ of SARS-CoV-2 Omicron VOC (B.1.1.529). One day after the viral challenge, we co-housed the index hamsters in pairs with contact hamsters for 4 days in clean cages. We euthanized the contact and index hamsters on day 4 of cohousing. Viral titers in the nasal tissue of the index and contact hamsters were used as primary endpoints. **B**, **C** Infectious viral particles in the nasal tissue of contact and index hamsters at day 5 after viral administration post-infection were measured by end point titration assay. The dotted line indicates LOD. Individual data points represent independent biological replicates taken from

separate animals; vertical bars show mean values with error bars representing SEM. Each dot represents an individual hamster. The analysis was performed using two-tailed Mann-Whitney U-test: ****$p < 0.0001$; ***$p < 0.001$; **$p < 0.01$; *$p < 0.05$; ns not significant. Asterisks indicate significance compared to the PBS-treated animals at each time point. Data presented as combined results from one independent experiment. Gender was tested as a variable with an equal number of male and female hamsters included in study. **A** and parts of (**B**, **C**): Created with BioRender.com released under a Creative Commons Attribution-NonCommercial-NoDerivs 4.0 International license (https://creativecommons.org/licenses/by-nc-nd/4.0/deed.en). Abbreviations: d2 Day 2, d5 Day 5, IN Intranasal. Number of animals used: $n = 8$–10/group. Source data are provided as a Source Data file.

Omicron VOC (BA.5) to vaccine-naïve hamsters, (3) it confers complete protection against SARS-CoV in mice, and (4) it elicits cross-reactivity Ig responses in both the serum and nasal compartment of NHPs.

Several next-generation SARS-CoV-2 and pan-sarbecovirus vaccines have been developed and these can be classified into three categories: (1) intranasal vaccines, (2) pan-sarbecovirus vaccines, and (3) dual-antigen vaccines, recognizing that the categories can be overlapping. Intranasal (and oral) vaccines for SARS-CoV-2 have been developed and translated to humans[57,58]. The vaccine based on a viral vector expressing the Wuhan S protein (ChAd-SARS-CoV-2-S) showed results similar to NanoSTING-S, demonstrating reduction but also variability in viral loads in the upper airways of K18-hACE2 mice challenged with chimeric viruses with spike genes corresponding to SARS-CoV-2 VOC (B.1.351 and B.1.1.28)[59]. Another vaccine candidate based on

adenovirus type 5–vectored SARS-CoV-2 Wuhan S protein was tested both using intranasal and oral administration, and both routes yielded only modest reduction in viral titers in hamsters[57]. The vaccine candidate was also tested in humans, but the program was subsequently abandoned[57]. A "prime and spike" vaccine (IM vaccination with the mRNA and an intranasal boost with the unadjuvanted Wuhan SARS-CoV-2 spike protein) showed efficacy comparable to dual dose mRNA vaccines against the ancestral SARS-CoV-2 (2020/USA-WA1) strain in mice: reduction (but not elimination) of viral titers in the lung and nostrils, and protection from weight loss in mouse models[60]. In a hamster transmission model, even a brief 4 h cohousing with an infected animal, allowed dual-dose vaccinated animals to pick up the infection. Although the vaccine was durable (challenge was performed 118 days after immunization), the efficacy was not tested against

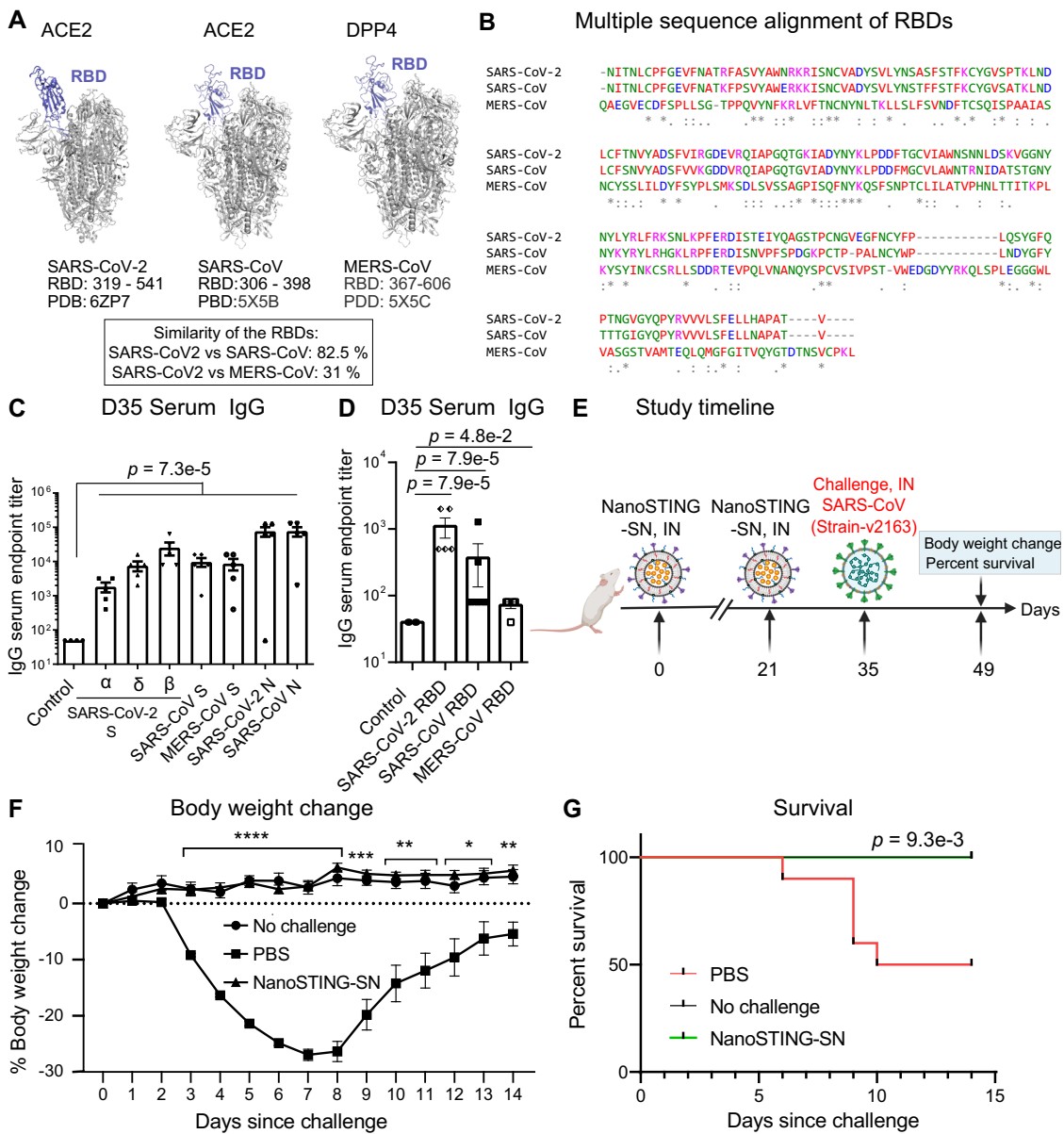

**Fig. 7 | Immunization of mice with NanoSTING-SN vaccine yields cross-reactive humoral immunity against betacoronaviruses and confers protection against SARS-CoV. A** 3D structure of SARS-CoV-2, SARS-CoV, and MERS-CoV spike proteins showing binding to respective receptors (PDB: 6ZP7, 5X5B, 5X5C). **B** Multiple sequence alignment of RBDs of SARS-CoV-2, SARS-CoV, and MERS-CoV spike (S) proteins. GenBank accession numbers are QHR63250.1 (SARS-CoV-2 S), AY278488.2 (SARS-CoV S), and AFS88936.1 (MERS-CoV S). **C, D** Humoral immune responses in the serum were evaluated using N and S protein-based IgG ELISA. **E** Experimental set up for SARS-CoV challenge studies in mice. We immunized mice (n = 10/group) intranasally with one dose of NanoSTING-SN on day 0 and a second dose on day 21 and challenged the mice intranasally with the SARS-CoV (v2163 strain) on day 35. Post-challenge, we monitored the animals for 14 days for changes in body weight and survival. **F** Percent body weight change of mice compared to the baseline at the indicated time intervals. **G** Percent survival of mice compared to the baseline at the indicated time intervals. Individual data points represent independent biological replicates taken from separate animals; vertical bars show mean values with error bars representing SEM. Each dot represents an individual mouse. For (**C, D**), the analysis was performed using two-tailed Mann-Whitney U-test: ****$p < 0.0001$; ***$p < 0.001$; **$p < 0.01$; *$p < 0.05$; ns not significant. For (**F**), the data

was compared via mixed-effects model for repeated measures analysis. Lines depict group mean body weight change from day 0; error bars represent SEM. For (**F**), the exact $p$ values comparing the NanoSTING-SN group to the Placebo group are Day 3: $p = 2.7e-7$, Day 4: $p = 1.9e-10$, Day 5: $p = 7.1e-12$, Day 6: $p < 1.0e-15$, Day 7: $p = 2.9e-13$, Day 8: $p = 1.2e-8$, Day 9: $p = 3.8e-4$, Day 10: $p = 8.0e-3$, Day 11: $p = 9.3e-3$, Day 12: $p = 2.2e-2$, Day 13: $p = 3.3e-2$, Day 14: $p = 7.4e-3$. Asterisks indicate significance compared to the PBS-treated animals at each time point. For (**G**), we compared survival percentages between NanoSTING-SN and PBS-treated animals using the Log-Rank Test (Mantel-Cox). Data presented as combined results from two independent experiments [**C, D**: Immunogenicity study with NanoSTING-SN, **E–G**: Challenge study with SARS-COV], each involving one independent animal experiment. Gender was not tested as a variable, and only female mice were used for the study (**C, D**). Gender was tested as a variable with an equal number of male and female mice included in the study (**E–G**). See also Supplementary Figs. 13, 14. **E** Created with BioRender.com released under a Creative Commons Attribution-NonCommercial-NoDerivs 4.0 International license (https://creativecommons.org/licenses/by-nc-nd/4.0/deed.en). Abbreviations: ACE2 angiotensin-converting enzyme 2, D35 Day 35, IN Intranasal. Number of animals used: **C, D**: n = 4–5/group, **E–G**: n = 10/group Source data are provided as a Source Data file.

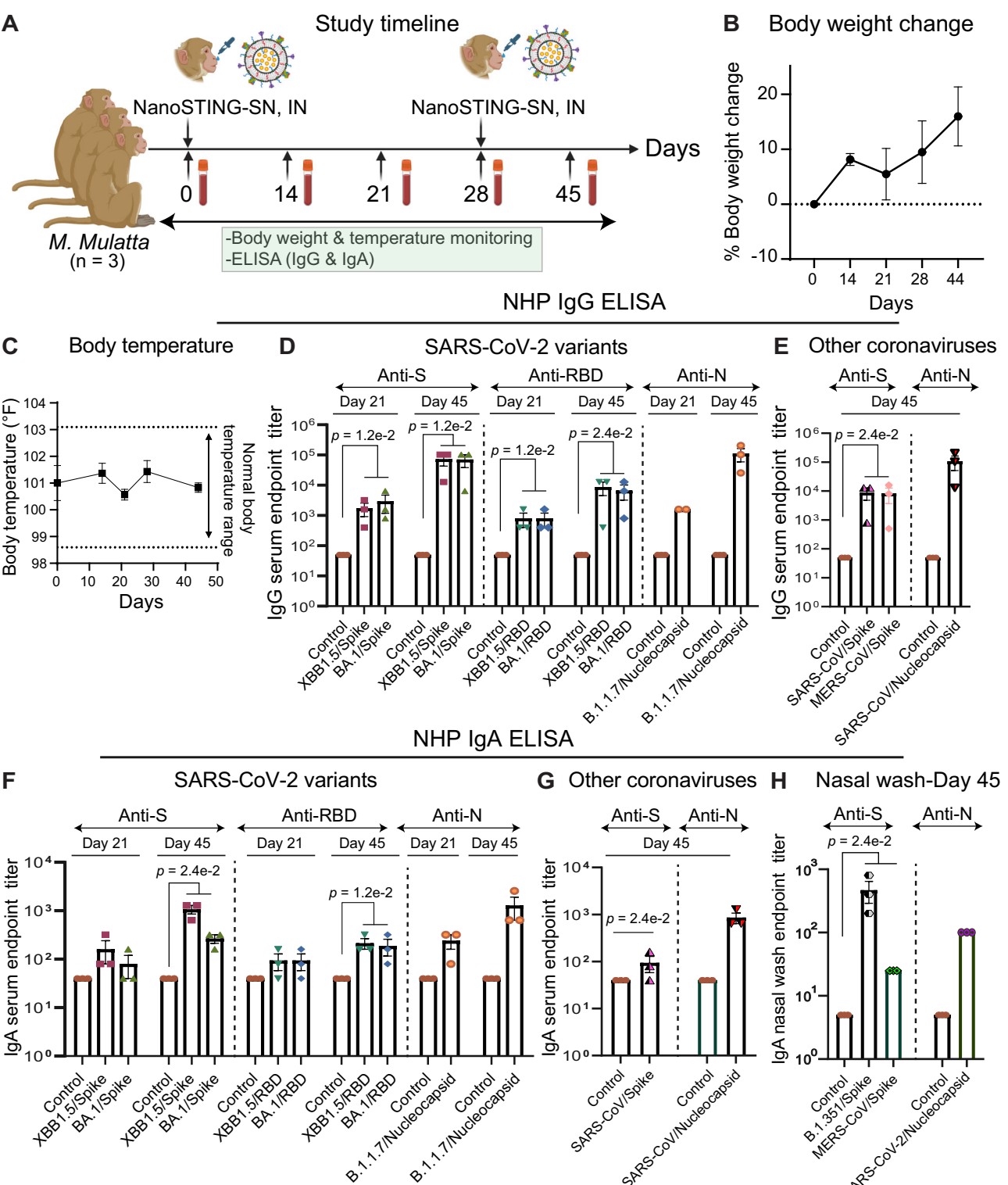

multiple strains of SARS-CoV-2, and protection against SARS-CoV required boosting with the intranasal spike derived from SARS-CoV[60]. From a vaccination perspective, having to continuously change the spike protein entails manufacturing of new proteins and, hence, is not scalable/translatable.

Pan-sarbecovirus vaccines designed based on either chimeric S proteins (mRNA based) or RBD-nanoparticles (protein-based) have been tested preclinically[61,62]. The S protein of sarbecoviruses comprises of three immunogenic domains: the N-terminal domain (NTD), the receptor binding domain (RBD), and the subunit 2 (S2). Synthetic

chimeric S proteins constructed by varying the NTD/RBD/S2 from different sarbecovirus were formulated as nucleoside-modified mRNA vaccines encapsulated in lipid nanoparticles (mRNA-LNP)[61]. To further facilitate cross-reactivity, mixtures of four separate such chimeric spikes were included in the vaccine and were administered IM in a prime boost regimen. These chimeric vaccines showed protection against mouse adapted strains of SARS-CoV and SARS-CoV-2 (Beta VOC) with robust protection from weight loss[61]. The use of K18-hACE2 mice and the lethality of the viruses in this mouse model precludes transmission studies to quantify the impact of the vaccines

**Fig. 8 | NanoSTING-SN confers durable humoral immunity in rhesus macaques.**
**A** Experimental setup: We administered two doses of the intranasal NanoSTING-SN vaccine (n = 3/group) 28 days apart to rhesus macaques. We collected the sera on days 0, 7, 14, 28, and 44 to evaluate humoral immune responses. We monitored the body weights of the animals every week after the immunization until the end of the study. Body weight change, body temperature change, and ELISA (IgG & IgA) were used as primary endpoints. Pre-immunization sera was used as control.
**B** Percent body weights change for the non-human primates. **C** Body temperature changes for the non-human primates. **D, E** Humoral immune responses in the serum were evaluated using N and S protein-based IgG ELISA. **F, G** Humoral immune responses in the serum were evaluated using N and S protein-based IgA ELISA.
**H** Humoral immune responses in the nasal washes were evaluated using N and S

protein-based IgA ELISA. Individual data points represent independent biological replicates taken from separate animals; vertical bars show mean values with error bar representing SEM. Each dot represents an individual animal. For (**D**–**H**), the analysis was performed using two-tailed Mann-Whitney U-test: ****$p < 0.0001$; ***$p < 0.001$; **$p < 0.01$; *$p < 0.05$; ns not significant. Data presented as combined results from one independent experiment. Two male and one female NHPs were used for the study. See also Supplementary Fig. 15. **A** Created with BioRender.com released under a Creative Commons Attribution-NonCommercial-NoDerivs 4.0 International license (https://creativecommons.org/licenses/by-nc-nd/4.0/deed.en). Abbreviations: IN Intranasal. Number of animals used: n = 3/group Source data are provided as a Source Data file.

on transmission. The dense and precise nanoscale organization of antigens is a feature of viruses considered essential for stimulating a robust humoral response[63]. The Spycatcher-spytag-based system leads to the display of 60mers of antigens, and this system was used to construct nanoparticles randomly displaying the RBDs of eight different sarbecoviruses (mosaic-8 RBD nanoparticles)[62]. Similar to the chimeric mRNA-LNPs described above, dual dose IM immunization of K18-hACE2 mice showed robust protection against weight loss when challenged with either SARS-CoV and SARS-CoV-2 (Beta) but formal transmission studies were not undertaken. Of note, the mosaic nanoparticles were tested in NHPs using a three-dose immunization regimen, and the vaccinated animals showed a ~100-fold reduction (but not elimination) of infectious virus in nasal swabs when challenged with the SARS-CoV-2 Delta VOC[62]. By contrast, dual dose immunization with NanoSTING-SN led to the complete elimination of the infectious virus in the nasal tissue in hamsters when challenged with the SARS-CoV-2 Delta VOC. Although the data are unavailable, intranasal immunization with mosaic-8 RBD nanoparticles adjuvanted with NanoSTING or even synthetic STING agonists like CF501 will provide an orthogonal formulation for pan-sarbecovirus vaccines[64].

Dual antigen vaccines targeting both the S and N proteins have been tested as either mRNA-LNP formulations or adenoviral vector vaccines[23,24]. The dual mRNA vaccine was administered IM in two doses and was tested against both SARS-CoV-2 Delta and Omicron VOCs in hamsters[23]. Consistent with our studies, using the two antigens showed additive protection, but consistent with IM immunization, the mRNA vaccines could not eliminate the virus in the nostrils. The adenoviral vector vaccine engineered to express both the S and N proteins was administered IM and tested against challenge by SARS-CoV-2 (2020/USA-WA1) strain in K18-hACE2 mice[24]. Comparisons of the S-only and the dual antigen vaccine formulations showed that while the S vaccine was sufficient to provide immunity in the lung, protection of the nervous system and the brain was only observed with the dual antigen vaccine. These observations complement our results with NanoSTING-SN and further reinforce the importance of multi-protein vaccines.

The ability to elicit multifactorial immunity in the nasal cavity has several implications for the design of vaccines targeting respiratory viruses. First, since respiratory viruses like SARS-CoV-2 can access the brain through the olfactory mucosa in the nasal cavity, immunity in this compartment can prevent viral seeding to the brain. This, in turn, can prevent the entire spectrum of neurological complications ranging from the immediate loss of smell and taste to long-term complications like stroke[65,66]. Second, eliminating the virus in the nasal cavity of vaccinated recipients reduces the chance of viral evolution leading to breakthrough disease, especially in the context of waning immunity[67]. Allowing the virus to persist is a risky experiment in viral evolution with likely tragic consequences[68]. Eradicating the SARS-CoV-2 viral reservoir in humans provides the only reasonable path to moving past the pandemic and the perpetual cycle of repeated booster vaccinations. Third, as the recent human data with Omicron infection in vaccinated hosts illustrates the importance of mucosal immunity in preventing

infection, and the identification of nasal antigen-specific IgA as a correlate of protection from infection helps the design of mucosal vaccines[53,54,69].

History provides a powerful example of the importance of vaccination to prevent infections, not just disease, and sets a clinical precedent. Similar to the current COVID-19 commercial vaccines, the first inactivated polio vaccine in 1955 successfully prevented disease but not infection. The availability of the oral polio vaccine starting in 1960 paved the way for eliminating infection and eradicating polio. The availability of multi-antigen mucosal vaccines provides a pathway for humanity to move past SARS-CoV-2 outbreaks.

In summary, we have developed and validated a multi-component intranasal NanoSTING-SN subunit vaccine candidate that directly eliminates transmission of highly transmissible variants and protects against multiple sarbecoviruses.

## Methods
### Preparation of NanoSTING, NanoSTING-S, NanoSTING-N, and NanoSTING-SN
The liposomes contained DPPC, DPPG, Cholesterol (Chol), and DPPE-PEG2000 (Avanti Polar lipids) in a molar ratio of 10:1:1:1. To prepare the liposomes, we mixed the lipids in $CH_3OH$ and $CHCl_3$. We used a vacuum rotary evaporator to evaporate them at 45 °C. We dried the resulting lipid thin film in a hood to remove residual organic solvent. Next, we added pre-warmed cGAMP (Medchem Express) solution (3 mg/mL in PBS buffer at pH 7.4) to hydrate the lipid film. We mixed the hydrated lipids for an additional 30 min at an elevated temperature of 65 °C and subjected them to freeze-thaw cycles. Using a Branson Sonicator (40 kHz), we next sonicated the mixture for 60 min and used Amicon Ultrafiltration units (MW cut off 10 kDa) to remove the free untrapped cGAMP. Finally, we used PBS buffer to wash the NanoSTING (liposomally encapsulated STINGa) three times. We measured the cGAMP concentration in the filtrates against a calibration curve of cGAMP at 260 nm using Take3 Micro-Volume absorbance analyzer of Cytation 5 (BioTek). We calculated the final concentration of cGAMP in NanoSTING and encapsulation efficiency by subtracting the concentration of free drug in the filtrate. To prepare NanoSTING adjuvanted subunit protein vaccine, we used a simple "mix and adsorb" approach. Briefly, (i) NanoSTING-S vaccine was prepared by gently mixing 10 μg of trimeric spike protein-B.1.351 (Acrobiosystems, #SPN-C52Hk) with 20 μg of NanoSTING. (ii) NanoSTING-N (Wuhan) (BEI, # NR-53797): Two different concentrations of the Nucleocapsid protein were taken: NanoSTING-N10 (10 μg of N protein) and NanoSTING-N20 (20 μg of N protein) were mixed separately with 20 μg of the NanoSTING. (iii) NanoSTING-N: 20 μg of nucleocapsid protein-B.1.17 (Acrobiosystems, #NUN-C52H8) was mixed with 20 μg of NanoSTING. (iv) NanoSTING-SN: 10 μg of trimeric spike protein-B.1.351 (Acrobiosystems, #SPN-C52Hk) and 20 μg of nucleocapsid protein-B.1.17 (Acrobiosystems, #NUN-C52H8) were mixed with 20 μg of NanoSTING. All the vaccines were left on ice for a minimum of 1 h with constant slow shaking on the rocker.

## Stability studies for the formulated vaccines

We stored the NanoSTING, NanoSTING-S, NanoSTING-N, and NanoSTING-SN at 4 °C for 6–9 months to check their stability. We measured the average hydrodynamic diameter and zeta potential of NanoSTING and all vaccine formulations using DLS and zeta sizer on Litesizer 500 (Anton Paar).

## Cell lines

THP-1 dual™ cells (NF-κB-SEAP IRF-Luc Reporter Monocytes) [Invivo-Gen, SanDiego, CA, thpd-nfis] were cultured in a humidified incubator at 37 °C and 5% CO$_2$ and grown in RPMI 1640, 2 mM L-glutamine, 25 mM HEPES, 10% heat-inactivated fetal bovine serum, 100 µg/ml Normocin™, Pen-Strep (100 U/ml-100 µg/ml). THP-1 dual cells were grown in the presence of respective selection agents [100 mg/mL zeocin (InvivoGen, #ant-zn-1)] and 10 mg/mL blasticidin (InvivoGen, #ant-bl-1)] every other passage to maintain positive selection of reporters.

## Cell stimulation experiments with luciferase reporter enzyme detection

We performed the THP-1 dual cell stimulation experiments using the manufacturer's instructions (InvivoGen, CA, USA). First, we seeded the cells in 96 well plate at $1 \times 10^5$ cells/well in 180 µL growth media. We then incubated the cells with 5 µg of NanoSTING at 37 °C for 24 h. To detect IRF activity, we collected 10 µL of culture supernatant/well at 12 h and 24 h and transferred it to a white (opaque) 96 well plate. Next, we read the plate on Cytation 7 (Cytation 7, Bio-Tek Instruments, Inc.) after adding 50 µL QUANTI-Luc™ (InvivoGen) substrate solution per well, followed by immediate luminescence measurement, which was given as relative light units (RLU).

## DNA binding assay

We performed the DNA binding studies as previously published[52]. To check the applicability of the assay for detecting DNA condensation, we used branched-chain PEI (Polyethylenimine) as a positive control (Sigma Chemical Co., St. Louis, MO # 408727). DiYO-1 (AAT Biore-quest #17579) and plasmid (pMB75.6)-DNA were mixed in equal volumes (in 20 mM HEPES, 100 mM NaCl, pH = 7.4) to achieve a final concentration of 400 nM DNA phosphate and 8 nM DiYO-1, respectively. The solution was left at room temperature (RT) for 5 h before use. Next, we added PEI at different concentrations (R = 0, 1, 2, 5 where R is the molar ratio of PEI Nitrogen to DNA phosphate) to DNA-DiYO-1 solution, mixed for 1 min, and left for 2 h to equilibrate. We measured the fluorescence intensity of the solution at excitation and emission wavelengths of 470 nm and 510 nm, respectively. We repeated the same procedure with SARS-CoV-2 N protein instead of PEI. To the DNA-DiYO-1 solution, we added the N protein at concentrations of 0.1 µM and 0.5 µM.

## Mice and immunization

All studies using animal experiments were reviewed and approved by the University of Houston (UH) IACUC. We purchased the female 7–9-week-old *BALB/c* mice from Charles River Laboratories (Strain code: 028). The mice were maintained within a Specific Pathogen Free (SPF) facility housed on ventilated racks within micro-isolation caging systems. Notably, the mice were not bred within the facility premises and were co-housed during the study. The housing facility for mice was under a 12:12-h light: dark cycle at temperatures 20–22 °C, humidity 40–50%. Before immunization, we anesthetized the groups of mice (n = 4–6/group) by intraperitoneal injection of ketamine (80 µg/g of body weight) and xylazine (6 µg/g of body weight). Then, we immunized the animals intranasally with (i) NanoSTING-S vaccine (ii) NanoSTING-N10 and NanoSTING-N20 (iii) NanoSTING-N (iv) NanoSTING-SN. All vaccines were freshly prepared.

## Body weight monitoring and sample collection

We monitored the body weight of the animals every 7 days until the end of the study after immunization. In addition, we collected the sera every week post-vaccination to detect the humoral immune response. We kept the blood at 25 °C for 10 min to facilitate clotting and centrifuged it for 5 min at 2000 × *g*. We collected the sera, stored it at −80 °C, and used it for ELISA. We harvested BALF, nasal wash, lung, and spleen at the end of the study, essentially as previously described[70,71]. We kept the sera and other biological fluids [with protease inhibitors (Roche, #11836153001)] at −80 °C for long-term storage. After dissociation, the splenocytes and lung cells were frozen in FBS + 10% DMSO and stored in the liquid nitrogen vapor phase until further use.

## Mouse ELISA

We determined the anti-N and anti-S antibody titers in serum or other biological fluids (BALF and nasal wash) using ELISA. Briefly, we coated 0.5 µg/ml S protein (α variant: Cat# NR-55311, BEI Resources, VA, USA; β variant: Cat# SPN-C52Hk, Acrobiosystems, DE, USA; γ variant: Cat# SPN-C52Hg, Acrobiosystems, δ variant: Cat# 10878-CV, R&D Systems, MN, USA; o variant: Cat# SPN-C52Hz, Acrobiosystems) and 1 µg/ml N protein (Sino Biological, PA, USA) onto ELISA plates (Corning, NY, USA) in PBS overnight at 4 °C or for 2 h at 37 °C. The plate was then blocked with PBS + 1% BSA (Fisher Scientific, PA, USA) + 0.1% Tween 20TM (Sigma-Aldrich, MD, USA) for 2 h at RT. After washing, we added the samples at different dilutions. We detected the captured antibodies using HRP-conjugated anti-mouse IgG (Jackson ImmunoResearch Laboratories, 1 in 6,000; PA, USA) and goat anti-mouse IgA biotin (Southern Biotech, 1: 5000; AL, USA). Streptavidin-HRP (Vector Laboratories, 1 in 2500, CA, USA) was used to detect the anti-IgA biotin antibodies. For BALF IgG ELISAs, antigens were coated onto plates at 0.5 µg/ml (S protein) and 2 µg/ml (N protein). We obtained the positive controls (anti-N and anti-S IgG) from Abeomics (CA, USA). We conducted three individual studies (NanoSTING-S, NanoSTING-N, and NanoSTING-SN), each with its own distinct control sets. We collected sera from the control animals upon completion of each study. The threshold for positivity was set at an optical density (450 nm) value of 0.05 or two times the negative control (PBS), whichever was higher. Endpoint titers were defined as the lowest dilution that was higher than the threshold for positivity.

## Mouse RBD ELISA

We evaluated RBD-specific serum IgG and IgA in mice using indirect ELISAs. We coated the plates with 0.5 µg/ml spike RBD proteins (α variant: Cat# NR-55277, BEI Resources, Manassas, VA; β variant: Cat# NR-55278, BEI Resources). We incubated the plates overnight at 4 °C or 2 h at 37 °C. The unbound protein was washed off by rinsing the wells twice with PBS + 0.05% Tween-20 (PBST). The remaining active protein binding sites on the plates were blocked off by incubating the plates with PBS + 1% BSA + 0.1% Tween-20 for 1 h at RT. After two additional washes with PBST, we added the serum samples serially diluted in PBST + 0.5% BSA. Endpoint titers were evaluated by adding serum samples in two-fold dilutions in duplicates. Following 1 h incubation with diluted serum, we washed the plates four times with PBST and added HRP-conjugated anti-mouse IgG (Cat# 115-035-166, Jackson ImmunoResearch Laboratories, 1: 5000; PA, USA) to detect RBD protein-specific IgG. To detect IgA in serum, we used Goat anti-mouse IgA biotin (Cat# 1040-05, Southern Biotech, 1: 5000; AL, USA). Streptavidin-HRP (Vector Laboratories, 1: 2500, CA, USA) was used to detect the anti-IgA biotin antibodies. We incubated the plates with detection antibodies for 1 h at room temperature. We washed the plates four times with PBST before adding 100 µl 1-Step™ TMB ELISA Substrate Solution (Cat# 34021, ThermoFisher, MA, USA). The plates were incubated with TMB for 30 min at room temperature, and we added 2 M H$_2$SO$_4$ to stop color development. Finally, we recorded optical density (OD) values using Cytation 7 (Biotek Instruments Inc).

## Processing of spleen and lungs for ELISPOT and flow cytometry

To isolate lung cells, we perfused the lung vasculature with 5 ml of 1 mM EDTA in PBS without $Ca^{2+}$, $Mg^{2+}$ and injected it into the right cardiac ventricle. Each lung was cut into 100–300 $mm^2$ pieces using a scalpel. We transferred the minced tissue to a tube containing 5 ml of digestion buffer containing collagenase D (2 mg/ml, Roche #11088858001) and DNase (0.125 mg/ml, Sigma #DN25) in 5 ml of RPMI for 1 h and 30 min at 37 °C in the water bath and vortexed after every 10 min. We disrupted the remaining intact tissue by passage (6–8 times) through a 21-gauge needle. Next, we added 500 µL of ice cold-stopping Buffer (1 × PBS, 0.1 M EDTA) to stop the reaction. We then removed tissue fragments and dead cells with a 40 µm disposable cell strainer (Falcon) and collected the cells after centrifugation at 400 × $g$. We then lysed the red blood cells (RBCs) by resuspending the cell pellet in 3 ml of ACK Lysing Buffer (Invitrogen) and incubated for 3 min at RT, followed by centrifugation at 400 × $g$. Then, we discarded the supernatants and resuspended the cell pellets in 5 ml of complete RPMI medium (Corning, NY, USA). Next, we collected the spleen in RPMI medium and homogenized them through a 40 µm cell strainer using the hard end of a syringe plunger. After that, we incubated splenocytes in 3 ml of ACK lysis buffer for 3 min at RT to remove RBCs, then passed through a 40 µm strainer to obtain a single-cell suspension. We counted the lung cells and splenocytes by the trypan blue exclusion method.

## ELISPOT

IFNγ and IL4 ELISpot assay was performed using Mouse IFNγ ELISPOT basic kit (ALP) and Mouse IL4 ELISPOT basic kit following the manufacturer's instructions (Mabtech, VA, USA). For cell activation control, we treated the cultures with 10 ng/ml phorbol 12-myristate 13-acetate (PMA) (Sigma, St. Louis, MI, USA) and 1 µg/ml of ionomycin (Sigma, St. Louis, MI, USA). We used the complete medium (RPMI supplemented with 10% FBS) as the negative control. We stimulated splenocytes and lung cells (3 × $10^5$) in vitro with either N-protein peptide pool (Miltenyi Biotec; #130-126-699, Germany) or an S-protein peptide pool (Genscript, # RP30020, USA) or S-protein (B.1.351) mutation peptide pool (Miltenyi Biotec, # 130-127-958, Germany) at a concentration of 1.5 µg/ml/peptide at 37 °C for 16–18 h in pre-coated ELISpot plate (MSIPS4W10 from Millipore) coated with AN18 IFNγ (1 µg/ml, Mabtech #3321-3-250;) and 11B11 IL4 (1 µg/ml, Mabtech #3311-3-250) coating antibody. The next day, we washed off the cells and developed the plates using biotinylated R4-6A2 anti-IFN-γ (Mabtech #3321-6-250) and BVD6-24G2 anti-IL4 (Mabtech #3311-6-250) detection antibody, respectively. Then, we washed the wells and treated them for 1 h at RT with 1:30,000 diluted Extravidin-ALP conjugate (Sigma, St. Louis, MI, USA). After washing, we developed the spots by adding 70 µL/well of BCIP/NBT-plus substrate (Mabtech #3650-10) to the wells. We incubated the plate for 20–30 min for color development and washed it with water. We quantified the spots using Cytation 7 (Bio-Tek Instruments, Inc.). Each spot corresponds to an individual cytokine-secreting cell. We showed the values as the background-subtracted average of measured triplicates.

## Cell surface staining, intracellular cytokine staining for flow cytometry

We stimulated the spleen and lung cells from immunized and control animals to detect nucleocapsid protein-specific CD8+ T cell responses with an N protein-peptide pool at a concentration of 1.5 µg/mL/peptide (Miltenyi Biotec; 130-126-699, Germany) at 37 °C for 16–18 h followed by the addition of Brefeldin A (5 µg/ml BD Biosciences #BD 555029) for the last 5 h of the incubation. We used 10 ng/ml PMA (Sigma, St. Louis, MI, USA) and 1 µg/ml ionomycin (Sigma, St. Louis, MI, USA) as the positive control. Stimulation without the peptides served as background control. We collected the cells and stained with Live/Dead Aqua (Thermo Fisher #L34965) in PBS, followed by Fc-receptor

blockade with anti-CD16/CD32 (Thermo Fisher #14-0161-85), and then stained for 30 min on ice with the following antibodies in flow cytometry staining buffer (FACS): anti-CD4 AF589 (clone GK1.5; Biolegend #100446), anti-CD8b (clone YTS156.7.7; Biolegend #126609), anti-CD69 (clone H1.2F3; Biolegend #104537), anti-CD137 (clone 1AH2; BD; # 40364), anti-CD45 (clone 30-F11; BD; #564279). We washed the cells twice with the FACS buffer. We then fixed them with 100 µL IC (intracellular) fixation buffer (eBioscience) for 30 min at RT. We permeabilized the cells for 10 min with 200 µL permeabilization buffer (BD Cytofix solution kit). We performed the intracellular staining using the antibodies Alexa Fluor 488 interferon (IFN) gamma (clone XMG1.2; BD; #557735) and Granzyme B (clone GB11; Biolegend; #515407) overnight at 4 °C. Next, we washed the cells with FACS buffer and analyzed them on LSR-Fortessa flow cytometer (BD Bioscience) using FlowJo™ software version 10.8 (Tree Star Inc, Ashland, OR, USA). We calculated the results as the total number of cytokine-positive cells with background subtracted. We optimized the amount of the antibodies by titration. See Supplementary Fig. 11A for the gating strategy.

## Viruses and biosafety

**Viruses.** We received the well-characterized challenge material (WCCM) from BEI Resources (Manassas, VA), which includes isolates of SARS-CoV-2 [NR-55612: SARS-Related Coronavirus 2 Isolate hCoV-19/USA/PHC658/2021 (Lineage B.1.617.2; Delta Variant), NR-58620: SARS-Related Coronavirus 2 Isolate hCoV-19/USA/COR-22-063113/2022 (Lineage BA.5; Omicron Variant)], NR-56462: SARS-Related Coronavirus 2 Isolate hCoV-19/USA/MD-HP20874/2021 (Lineage B.1.1.529; Omicron Variant) and SARS-CoV (NR-15418 SARS coronavirus Urbani v2163). We amplified the viruses in Vero E6 cells to create working stocks of the virus. The virus was adapted to mice by four serial passages in the lungs of mice and plaque purified at Utah State University (USU).

**Biosafety and Ethics.** The animal experiments at USU were conducted in accordance with an approved protocol by the Institutional Animal Care and Use Committee of USU. The work was performed in the AAALAC-accredited LARC of the university in accordance with the National Institutes of Health Guide for the Care and Use of Laboratory Animals (8th edition; 2011).

## Viral challenge studies in animals

**Animals:** For SARS-CoV-2 animal studies completed at USU, 6–10-week-old male and female golden Syrian hamsters (*Mesocricetus auratus*) were purchased from Charles River Laboratories (Strain code: 049) and housed in the ABSL-3 animal space within the LARC. All hamsters included in the study were carefully matched for age. The hamsters were not bred on-site. All hamsters were singly housed while at the facility.

**Infection of animals.** Hamsters were anesthetized with isoflurane and infected by intranasal instillation of $1 \times 10^{4.5}$ $CCID_{50}$ (cell culture infectious dose 50%) of SARS-CoV-2 in a 100 µl volume.

**Titration of tissue samples:** Lung tissue and nasal tissue samples from hamsters were homogenized using a bead-mill homogenizer using minimum essential media. Homogenized tissue samples were serially diluted in a test medium, and the virus was quantified using an endpoint dilution assay on Vero E6 cells for SARS-CoV-2. A 50% cell culture infectious dose was determined using the Reed-Muench equation[72].

## Transmission studies

We immunized hamsters with a dual dose of the intranasal NanoSTING-SN vaccine or PBS, 5 weeks (day-21) and 2 weeks (Day 0) prior to infection with ~3 × $10^4$ $CCID_{50}$ of SARS-CoV-2 Omicron VOC (BA.5) [day 14]. One day after the viral challenge, we co-housed the index hamsters in pairs with contact hamsters for 4 days in clean cages. We euthanized the index and contact hamsters on day 4 and day 5 of cohousing. Viral

titers in the lungs and nasal tissue of the index and contact hamsters were used as primary endpoints. We performed another study with another strain of SARS-CoV-2 Omicron VOC (B.1.1.529). We immunized hamsters with a single dose of the intranasal NanoSTING-SN vaccine or PBS, 3 weeks prior to infection with ~3 × 104 CCID$_{50}$ of SARS-CoV-2 Omicron VOC (B.1.1.529). One day after the viral challenge, we co-housed the index hamsters in pairs with contact hamsters for 4 days in clean cages. We euthanized the contact and index hamsters on day 4 of cohousing. Viral titers in the nasal tissue of the index and contact hamsters were used as primary endpoints.

### Histopathology

Lungs of the Syrian golden hamsters were fixed in 10% neural buffered formalin overnight and then processed for paraffin embedding. The 4-µm sections were stained with hematoxylin and eosin for histopathological examinations. We used integrated scoring rubric for evaluating the pathology score[73]. The scoring method in the reference was modified from a 0–3 to a 0–4 score with 1 = 1–25%, 2 = 26–50%; 3 = 51–75; 4 = 76–100%, so with the three criteria mentioned in the reference will yield a score for an animal ranging from 0–12. This scoring also takes into account the degeneration/necrosis of the bronchial epithelium/alveolar epithelium. A board-certified pathologist evaluated the sections.

### Quantitative modeling

To quantify the kinetics of SARS-CoV-2 infection in the upper respiratory tract (URT) upon N, S, and N + S immunization, we modified a previously described innate immune model[46]. We added appropriate parameters to account for de-novo blocking and T-cell killing, as shown in Fig. 2A. The mean population parameter values and initial values were from prior publication[46]. We solved the system of ordinary differential equations (ODEs) for different S and N response efficiencies using the ODE45 function in MATLAB 2018b. A sample MATLAB code for solving the system of equations has been provided in Sup Note 1. There are some limitations to our model. This model does not consider the effect of interferons that are known to suppress viral production rate. Under these conditions, the rate constant $\pi$, for viral production should reduce as the infection progresses and this factor is especially important when the infection persists >7 days[46]. Second, our model in Supplementary Fig. 5C predicts that the number of viral particles is between 1 and 100 which is below the experimental detection limit and hence cannot be confirmed experimentally.

### Immunogenicity studies in rhesus macaques (RM's) and their monitoring

Experiments with rhesus macaques (*M. mulatta*) were reviewed and approved by UH IACUC. Three healthy rhesus macaques (RM's) of Indian origin, between 4 and 11 years of age and 4–12 kg in weight) were used. The RM's were acquired from Washington University School of Medicine, Division of Comparative Medicine C/O Dr. Chad B Faulkner; 660S. Euclid Ave., Box 8061; St. Louis, MO 63110 and Keeling Center for Comparative Medicine and Research, MD Anderson Cancer Center, Bastrop, TX. We used three RM's for the study. Two of them were males, and one was female. All the animals were single-housed. We administered two doses of the intranasal NanoSTING-SN vaccine (*n* = 3/group), 28 days apart to RM's The animals were monitored until day 44 for changes in body weight, attitude, appetite, body temperature (via a rectal thermometer). For evaluating humoral immune responses, we collected the sera and nasal washes on day 0, 7, 14, 28, and 44. We kept the blood at 25 °C for 10 min to facilitate clotting and centrifuged it for 5 min at 2000 × *g*. We collected the sera, stored it at −80 °C, and used it for ELISA. We kept the sera and nasal wash fluid [with protease inhibitors (Roche, #11836153001] at −80 °C for long-term storage.

**NHP ELISA**. For NHP serum IgG and IgA ELISA, we coated the plate overnight with 0.5 µg/ml S protein and 1 µg/ml N protein (Acrobiosystems, DE, USA). The MERS spike protein (BEI resources, VA, USA) was also coated at 0.5 µg/ml. Subsequent blocking and wash steps were performed similarly to the mouse IgG and IgA ELISAs. Serum and nasal wash samples from NHPs were added at different dilutions and incubated for 1 h at room temperature. Mouse anti-monkey IgG HRP (Southern Biotech, 1: 5000; AL, USA) was used to detect IgG in serum and nasal wash samples. IgA was detected using goat anti-monkey IgA HRP from Exalpha (MA, UK) at 1:5000 dilution. Pre-immunization sera was used as control for ELISA. The threshold for positivity was set at an optical density (450 nm) value of 0.05 or two times the negative control (PBS), whichever was higher. Endpoint titers were defined as the lowest dilution that was higher than the threshold for positivity.

**NHP RBD ELISA**. We evaluated RBD-specific serum IgG and IgA in NHPs using indirect ELISAs. We coated the plates with 0.5 µg/ml spike RBD proteins (BA.1 variant: Cat# NR- SPD-C522j, ACROBiosystems, DE, USA); (XBB1.5 variant: Cat# SPD-C5242, ACROBiosystems, DE, USA). We incubated the plates overnight at 4 °C or 2 h at 37 °C. The unbound protein was washed off by rinsing the wells twice with PBS + 0.05% Tween-20 (PBST). The remaining active protein binding sites on the plates were blocked off by incubating the plates with PBS + 1% BSA + 0.1% Tween-20 for 1 h at room temperature. After two additional washes with PBST, we added the serum samples serially diluted in PBST + 0.5% BSA. Endpoint titers were evaluated by adding serum samples in two-fold dilutions in duplicates. Following 1 h of incubation at RT with diluted serum, we washed the plates four times with PBST and added HRP-conjugated anti-human IgG (Cat# 6200-05, Southern Biotech, 1: 80000); AL, USA to detect RBD protein-specific IgG. For the detection of IgA in serum, we used mouse anti-monkey IgA biotin (Cat# MCA2553B, BioRad, 1: 10000; CA, USA). We incubated the plates with detection antibodies for 1 h at room temperature. We washed the plates four times with PBST and added Streptavidin-HRP (Vector Laboratories, 1: 5000, CA, USA) to detect the anti-IgA biotin antibodies. Finally, we washed the plates four times with PBST before adding 100 µl 1-Step™ TMB ELISA Substrate Solution (Cat# 34021, ThermoFisher, MA, USA). The plates were incubated with TMB for 30 min at RT, and we added 2 M H$_2$SO$_4$ to stop color development. We recorded optical density (OD) values using Cytation 7 (Biotek Instruments Inc).

### Statistics and reproducibility

Statistical significance was assigned when *P* values were <0.05 using GraphPad Prism (v6.07). Tests, number of animals (n), mean values, statistical comparison groups, and the statistical test used are indicated in the figure legends. No statistical methods were used to pre-determine sample sizes for the in-vitro and animal studies. Sample size was determined based on similar studies in this field. Animals were randomly divided into experimental groups. When applicable, technical repeats are specified for each experiment in the figure legends wherever applicable. No data was excluded from the analyses. Animal studies were performed in biological triplicates or more, as indicated in the figure legends. Reproducibility between animals in NanoSTING, NanoSTING-N, and NanoSTING-SN and PBS groups is shown in the results and figure legends. The researchers were not blinded to allocation during experiments and outcome assessment. Data collection and analysis were not performed blind to the conditions of the experiments. The pathologists performing the histopathological analysis were blinded to treatment. The adjuvant was manufactured at UH, and the adjuvant/protein were shipped to USU. USU performed the vaccine formulation for the challenge experiments and immunized and challenged the animals. Further information on research design is available in the Nature Research Reporting Summary linked to this article.

**Reporting summary**

Further information on research design is available in the Nature Portfolio Reporting Summary linked to this article.

## Data availability

All data are included in the source data file or available from the authors, as are unique reagents used in this article. The raw numbers for charts and graphs are available in the Source Data file whenever possible. All material and experimental data requests should be directed to the corresponding author, Navin Varadarajan. Source data are provided with this paper.

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

## Acknowledgements

This publication was supported by the NIH (R01GM143243), AuraVax Therapeutics, and Owens Foundation. X.L. acknowledges partial funding support from the National Cancer Institute (NIH R15CA182769, P20CA221731, P20CA221696) and CPRIT (RP150656). The following reagents were produced under HHN272201400008C and obtained through BEI Resources, NIAID, NIH: Spike Glycoprotein (Stabilized) from SARS-Related Coronavirus 2, Wuhan-Hu-1, Recombinant from Baculovirus, NR-52308; and Spike Glycoprotein Receptor Binding Domain (RBD) from SARS Related Coronavirus 2 (multiple variants), NR-55612: SARS-Related Coronavirus 2 Isolate hCoV-19/USA/PHC658/2021 (Lineage B.1.617.2; Delta Variant): This reagent was obtained through BEI Resources, NIAID, NIH: SARS-Related Coronavirus 2, Isolate hCoV-19/USA/PHC658/2021 (Lineage B.1.617.2; Delta Variant) (WCCM), NR-55612, contributed by Dr. Richard Webby and Dr. Anami Patel. NR-58620: SARS-Related Coronavirus 2 Isolate hCoV-19/USA/COR-22-063113/2022 Lineage BA.5; Omicron Variant: This reagent was obtained through BEI Resources, NIAID, NIH: SARS-Related Coronavirus 2, Isolate hCoV-19/USA/COR-22-063113/2022 (Lineage BA.5; Omicron Variant) in VeroTMPRSS2-ACE2 Cells (WCCM), NR-58620, contributed by

Dr. Richard J. Webby. NR-56462: SARS-Related Coronavirus 2 Isolate hCoV-19/USA/MD-HP20874/2021 (Lineage B.1.1.529; Omicron Variant) was obtained through BEI Resources, NIAID, NIH: SARS-Related Coronavirus 2, Isolate hCoV-19/USA/MDHP20874/2021 (Lineage B.1.1.529; Omicron Variant) (WCCM), NR-56462, contributed by Andrew S. Pekosz. NR-15418: SARS coronavirus Urbani v2163: Severe acute respiratory syndrome coronavirus (SARS-CoV), strain Urbani (200300592), was obtained from the Centers for Disease Control and Prevention (CDC, Atlanta, GA) and routinely passaged in Vero-76 cells. This virus was adapted through 25 serial passages in the lungs of mice and plaque-purified for use in mouse infections[74]. Schematics were made using Adobe Illustrator, Microsoft PowerPoint, and Biorender via full license.

## Author contributions

N.V. conceived the study. N.V., A.L., A.S., L.J.N.C., M.S., and B.H. designed the study. A.L., A.S., K.M.S.R.S., M.K., M.M.P., A.D., R.K., K.R., A.R., S.M., B.H., C.M.S., V.E.D., M.S., N.B.P., and X.L. performed experiments. A.L., A.S., K.M.S.R.S., M.K., B.H., X.L., and N.V. analyzed the data. M.K. performed modeling. N.V., A.L. drafted the manuscript, and all authors contributed to the review and editing of the manuscript.

## Competing interests

UH has filed a provisional patent based on the findings of this study. N.V. and L.J.N.C. are co-founders of AuraVax Therapeutics and CellChorus. The remaining authors declare no competing interests.

## Ethical approval

The mouse, hamster, and NHP studies were performed under the study protocol (PROTO2020000019, PROTO202100006, PROTO202100049, PROTO202200025), as approved by the Institutional Animal Care and Use Committee in University of Houston. The animal experiments at USU were conducted in accordance with an approved protocol by the Institutional Animal Care and Use Committee of Utah State University. The work was performed in the AAALAC-accredited LARC of the university in accordance with the National Institutes of Health Guide for the Care and Use of Laboratory Animals (8th edition; 2011).
