## [Peer Review File · Nature Communications]

Multi-antigen intranasal vaccine protects against challenge with sarbecoviruses and prevents transmission in hamstersReviewers' Comments:

Reviewer #1:

Remarks to the Author:

In this manuscript, Leekha et al. developed an intranasal vaccine that can protect against challenge by SARS-CoV-2 delta and omicron VOCs. This vaccine is consisting of spike and nucleocapsid proteins as immunogens, adjuvanted with liposomes encapsulated STING agonists. This NanoSTING-SN can induce both humoral and cellular immunities in the mucosal lymphoid tissues that leads to the rapid elimination of viral loads in both the lungs and the nostril following live virus challenges.

This is an informative study with detailed analysis, comparison and characterization using a variety of experimental methods and techniques. This work also holds great interest for intranasal vaccine development. The background and significance of this study is clear. The rationale and logic of the paper need to be improved. The context fits well to the scope of the field in this journal, I recommend this paper to be published in Nat. Commun. after major revisions.

Major comment #1:

The authors claimed the NanoSTING-SN can induce sterilizing immunity, which requires neutralizing antibodies elicited upon vaccination. However, the authors did not perform the neutralization assay in any serum samples from mice, hamsters and NHPs. In their previous study [ref: An, X. et al. Single-dose intranasal vaccination elicits systemic and mucosal immunity against SARS-CoV-2. *iScience* 24, 103037 (2021)], they measured the antibody response using a pseudovirus neutralization assay. The binding assay of measuring the IgG or IgA responses cannot represent the neutralizing antibody response against SARS-CoV-2 VOCs. The pseudovirus neutralization assay needs to be performed in this study.

Major comment #2:

The authors mentioned that "antibodies against the N protein not being neutralizing as this protein is unassociated with viral entry" and "this concern of enhanced respiratory disease mediated by Th2 responses has shifted the focus away from the SARS-CoV-2 N protein-based vaccines despite the potential for protective T-cell responses". As the authors reviewed in the introduction section, N protein does not generate neutralizing antibody but induce T cell mediated cellular immunity. What is the rationale to include this nucleocapsid (N) protein component in the vaccine formulation if the authors aim sterilizing immunity, which is based on neutralizing antibody responses.

Major comment #3:

The authors utilized the recombinant trimeric S-protein based on the SARS-CoV-2 B.1.351 (beta VOC) as the vaccine immunogen. However, beta variant was rapidly replaced by delta and later omicron. What is the rationale of using beta VOC as an immunogen? How about Wildtype Wuhan stain, or other VOCs as immunogens? Are the authors expecting the same immune responses?

Major comment #4:

For the NanoSTING-S, how did the authors measure the amount of S on a liposome? For the NanoSTING-N, how did the authors conjugate or absorb the N proteins on NanoSTING liposome? It is unclear the amount of N on a liposome. Did the authors purified the NanoSTING-SN? How did the authors normalize the injection dose for NanoSTING-S, NanoSTING-N and NanoSTING-SN? The authors need to include more detailed characterization in the sections of method and results.

Major comment #5:

The authors mentioned that "they stimulated the spleen and lung cells with a pool of peptides containing mutations (B.1.351) in the S protein that differs from the Wuhan S protein. They observed a significant Th1 responses against these mutation-specific S peptides confirming a broad T-cell response that targets both the conserved regions and the mutated regions of the S protein". Why the

authors compared with the Wuhan stain if they used beta VOC as immunogen? How did this result confirm the broad T cell response if they used the matched stain?

Major comment #6:

The authors evaluated the durability of the response upon vaccination at five months after immunization (Figure S4). However, the durability or longevity of the antibody response need to be evaluated in a figure. For example, IgG at 1 month vs. IgG at 5 months. Then the statistical analysis needs to be performed to evaluate the antibody responses over time.

Major comment #7:

The logic of this experimental design needs to be clarified. The immune responses induced by NanoSTING-N should be moved in front of the NanoSTING-SN combined.

Major comment #8:

The authors reported a single dose of NanoSTING-SN significantly reduces direct transmission of SARS-CoV-2 Omicron VOC. Did the authors perform any immunological analysis related to the immune responses after a single dose of immunization? Any survival or weight loss data? Vaccine induced humoral and cellular immunities?

Other minor comments:

1. The authors need to write the full name when it appears for the first time in the text but not the abbreviation, such as NanoSTING-SN.

Reviewer #2:

Remarks to the Author:

This manuscript intends to demonstrate that mucosal multi-antigen vaccines present a pathway to end transmission of respiratory viruses, and that presenting multiple viral proteins as immunogens may be advantageous in engineering pan-sarbecovirus vaccines. Unfortunately, this reviewer is not convinced by the results presented on either point. I don't see a reduction in transmission rate of infection of directly exposed animals in their data and the second item also appears to be rather weak.

1. The authors listed anti-RBD ELISA titers as a surrogate for neutralization without reference or validation of this position.

2. SARS-CoV-2 isolates are not referenced properly in terms of source (BEI number, etc).

3. It looks like the authors recycled data from their PBS mock control and baseline control hamsters in Figures 1, 3, and 4 but did not clearly mention this in the text. If the intention is to compare immunogenicity and efficacy between NanoSting-S and NanoSting-SN, why not plot the two groups of data side by side? As a matter of fact, based on Fig. 1N and Fig. 3M, the NanoSting-S appears to reduce lung pathogen to a higher extent than NanoSting-SN.

4. I am not clear how Fig. 2 adds anything.

5. Figure 4C: Y-axis is log₂ instead of log₁₀ like the other figures, is there a particular reason?

6. Figure 5B: The transmission observed by the authors is 75% at d2 and 100% at d5 for both vaccinated and mock-vaccinated hamsters, and although a case can be made for reduced disease in contact hamsters the amount of "shed" virus in the nasal washes does not seem to be significantly reduced in the vaccinated hamsters. "Ending transmission" is clearly an overstatement.

7. The authors may perform a comparative study between intramuscular and intranasal vaccination

using their vaccine candidates to demonstrate the advantage of the latter.

Reviewer #3:

Remarks to the Author:

I would like to congratulate you to this very nicely written manuscript. It is very pleasing to see what kind of central role the mathematical modelling part holds in this manuscript. This is encouraging for interdisciplinary work and shows how experimental work can benefit from mathematical models. A possible vaccine strategy against S and N proteins of SARS-COV2 was mathematically tested and revealed that a synergetic effect would result in the best outcome - reducing viral titres of SARS-COV2. An intranasal S+N-vaccine has been designed and then tested in several animal models. The mathematical "prediction" could be approved, and the vaccine efficiency has been further tested with positive outcome regarding transmission and cross-reactivity.

Major comments:

- Is there enough experimental data available, so that the model (derived on human data) can be transferred to the related animal model? It would be nice to see if an animal derived model could capture the experimental data. If this is not possible, please discuss.
- The mathematical model suggests (according to Figure S5C) a lower viral titre for S+N vaccination, but the persistence of the viral load seems to be prolonged. Can that also be observed experimentally? Can you explain the basis of this result?

Minor comments:

- Line 128: Why was not also tested for beta IgA?
- Line 129/130: It is not intuitive how IFN γ and IL4 translate to Th1/TC1 responses.
- Line 174: The N protein is chosen for what? Context is not completely clear; it might help to switch this sentence with the following.
- Lines 177-179: This sentence is very confusing and requires further explanation. As far as I am aware there is no parameter fitting performed in this manuscript – rather, you explored the parameter space manually. Of which infected patients are we talking here otherwise?
- Lines 179-187: It is not clear from the text if this part is based on S-protein only? I initially thought, it's the response against S+N.
- Line 683: N+S (otherwise confusion with not significant)
- Figure1: SEM – standard error on the mean? Please be specific and introduce abbreviation.
- Figure2: Is it possible to separate the figure in 4 sections (I: no vaccination, II: S only, III: N only, IV: S+N)? It would be easier to refer to in the main text as well as more understandable for the reader.
- Figure2A: The schematic illustration requires a proper description in the legend.
- Figure2: AUC – area under the curve
- Figure2C: Does omega vary between 0 and 1 or 0 and 10?
- Figure2D: Why is now no longer reduction in viral AUC used as a measure?
- Figure3A: What is the reason that the protocol changed to euthanize the animals now on day 51 instead on day 28 (previous study)?
- Figure3B: Why was not tested against omicron IgG?
- Figure4B: It is not clear against which strains was tested.
- FigureS5: The model output in panel A looks anomalous for $\gamma=0.8$ (blue curve). Can you explain this? Does it eventually decline? Extension of the x-axis would be appropriate to see the full AUC.
- FigureS5: A more detailed legend is required.
- FigureS5: Replace (D) by (C) in legend.

Reviewers' comments:

Reviewer #1 (Remarks to the Author):

In this manuscript, Leekha et al. developed an intranasal vaccine that can protect against challenge by SARS-CoV-2 delta and omicron VOCs. This vaccine is consisting of spike and nucleocapsid proteins as immunogens, adjuvanted with liposomes encapsulated STING agonists. This NanoSTING-SN can induce both humoral and cellular immunities in the mucosal lymphoid tissues that leads to the rapid elimination of viral loads in both the lungs and the nostril following live virus challenges.

This is an informative study with detailed analysis, comparison and characterization using a variety of experimental methods and techniques. This work also holds great interest for intranasal vaccine development. The background and significance of this study is clear. The rationale and logic of the paper need to be improved. The context fits well to the scope of the field in this journal, I recommend this paper to be published in Nat. Commun. after major revisions.

Response: We thank the reviewer for their many positive comments.

Major comment #1:

The authors claimed the NanoSTING-SN can induce sterilizing immunity, which requires neutralizing antibodies elicited upon vaccination. However, the authors did not perform the neutralization assay in any serum samples from mice, hamsters and NHPs. In their previous study [ref: An, X. et al. Single-dose intranasal vaccination elicits systemic and mucosal immunity against SARS-CoV-2. iScience 24, 103037 (2021)], they measured the antibody response using a pseudovirus neutralization assay. The binding assay of measuring the IgG or IgA responses cannot represent the neutralizing antibody response against SARS-CoV-2 VOCs. The pseudovirus neutralization assay needs to be performed in this study.

Response: Although we have performed pseudovirus neutralization (PsVN) assays in our previous publication, these were needed since we did not perform animal challenge experiments. In the current paper, we have used the ELISAs against RBDs as a surrogate for PsVN assays since this allows us to test the sera against many different SARS-CoV-2 variants (e.g. see PMIDs: 33399823 and 34971839). In the current manuscript, we have complemented the RBD ELISAs with multiple animal challenge experiments and have directly measured infectious virus upon challenge of vaccinated animals.

Major comment #2:

The authors mentioned that “antibodies against the N protein not being neutralizing as this protein is unassociated with viral entry” and “this concern of enhanced respiratory disease mediated by Th2 responses has shifted the focus away from the SARS-CoV-2 N protein-based vaccines despite the potential for protective T-cell responses”. As the authors reviewed in the introduction section, N protein does not generate neutralizing antibody but induce T cell mediated cellular immunity. What is the rationale to include this nucleocapsid (N) protein component in the vaccine formulation if the authors aim sterilizing immunity, which is based on neutralizing antibody responses.

Response: Thank you for your insights. Our study demonstrated that intranasal vaccination with NanoSTING-N induced strong Th1/Tc1 responses without any evidence of Th2 responses. These findings highlight the vaccine's ability to elicit protective T-cell immunity without the concern of Th2-mediated respiratory issues. While the N protein may not generate neutralizing antibodies, our study emphasizes its role in fostering T-cell mediated cellular immunity and complements the antibody response against the spike protein. Furthermore, studies investigating the role of immune responses against the N protein have demonstrated that antibody responses against the N protein can induce antibody dependent cell mediated cytotoxicity (ADCC) by natural killer cells (PMID: 36219482). Although we have not investigated this phenomenon, our rationale for including the N protein relies on its ability to induce a multifactorial immune response.

Major comment #3:

The authors utilized the recombinant trimeric S-protein based on the SARS-CoV-2 B.1.351 (beta VOC) as the vaccine immunogen. However, beta variant was rapidly replaced by delta and later omicron. What is the rationale of using beta VOC as an immunogen? How about Wildtype Wuhan stain, or other VOCs as immunogens? Are the authors expecting the same immune responses?

Response: Although the spike immunogen in the current manuscript is based on the SARS-CoV-2 B.1.351 (Beta VOC), we have demonstrated the efficacy of this immunogen to protect against the SARS-CoV2 BA.5 (Omicron VOC). The beta variant prevailed during vaccine formulation, prompting its selection for studying variant-specific immune responses. Despite its replacement by subsequent variants, our study revealed promising cross-reactive immune responses elicited by the spike protein from Beta VOC, suggesting potential broad protection against different VOCs.

Major comment #4:

For the NanoSTING-S, how did the authors measure the amount of S on a liposome? For the NanoSTING-N, how did the authors conjugate or absorb the N proteins on NanoSTING liposome? It is unclear the amount of N on a liposome. Did the authors purify the NanoSTING-SN? How did the authors normalize the injection dose for NanoSTING-S, NanoSTING-N and NanoSTING-SN? The authors need to include more detailed characterization in the sections of method and results.

Response: As outlined in our previous paper highlighted by the reviewer (PMID: 34462731), we performed ELISA-based quantification to demonstrate that the spike protein was absorbed onto the liposomes. Broadly, we emphasize our formulation relies on a mix and immunize approach as outlined in the methods section. Consistently across all the vaccines, the dose of protein is the total protein added to the formulation regardless of the fraction absorbed onto the liposomes.

Major comment #5:

The authors mentioned that “they stimulated the spleen and lung cells with a pool of peptides containing mutations (B.1.351) in the S protein that differs from the Wuhan S protein. They observed a significant Th1 responses against these mutation-specific S peptides confirming a broad T-cell response that targets both the conserved regions and the mutated regions of the S protein”. Why the authors compared with the Wuhan stain if they used beta VOC as immunogen? How did this result confirm the broad T cell response if they used the matched stain?

Response: The comparison with the SARS-CoV-2 Wuhan strain highlighted the specificity of T-cell responses against the mutation-specific peptides of the B.1.351 variation. By testing peptides from both the ancestral strain and the mutated peptides we are able to demonstrate that the T-cell responses target both the conserved and mutated regions.

Major comment #6:

The authors evaluated the durability of the response upon vaccination at five months after immunization (Figure S4). However, the durability or longevity of the antibody response need to be evaluated in a figure. For example, IgG at 1 month vs. IgG at 5 months. Then the statistical analysis needs to be performed to evaluate the antibody responses over time.

Response: We thank the reviewer for this suggestion, we have updated the supplementary figure showing these comparisons. Additionally, we have also evaluated the durability of the antibody response in NHPs at 6 months (**Figure S15**)

Updated figures (Pg. 39 & Pg. 51): Figures S4 and S15

Added text (Pg., Line:355-356): “Importantly, the IgG responses against both the S and N proteins did not reduce significantly over 6 months in the immunized NHPs (**Figure S15**)”.

Major comment #7:

The logic of this experimental design needs to be clarified. The immune responses induced by NanoSTING-N should be moved in front of the NanoSTING-SN combined.

Response: We appreciate the context of your suggestion to present NanoSTING-N immune responses before NanoSTING-SN in order to clearly define its role. However, the primary goal of our experiment was to demonstrate the combined efficacy of NanoSTING-SN and the sterilizing immunity it conferred.

Major comment #8:

The authors reported a single dose of NanoSTING-SN significantly reduces direct transmission of SARS-CoV-2 Omicron VOC. Did the authors perform any immunological analysis related to the immune responses after a single dose of immunization? Any survival or weight loss data? Vaccine induced humoral and cellular immunities?

Response: We have updated data where we have shown administration of two doses of the vaccine (**Figure 5**). These results demonstrated that NanoSTING-SN is highly effective at preventing the transmission of the SARS-CoV-2 Omicron VOC (BA.5). The animals showed no weight loss and 100% survival, consistent with other studies that used Omicron VOCs (e.g. PMIDs: 35737809 and 35313451).

Added figure (Pg. 30): Figure 5

Other minor comments:

1. The authors need to write the full name when it appears for the first time in the text but not the abbreviation, such as NanoSTING-SN.

Response: We have made the changes in the manuscript file.

Reviewer #2 (Remarks to the Author):

This manuscript intends to demonstrate that mucosal multi-antigen vaccines present a pathway to end transmission of respiratory viruses, and that presenting multiple viral proteins as immunogens may be advantageous in engineering pan-sarbecovirus vaccines. Unfortunately, this reviewer is not convinced by the results presented on either point. I don't see a reduction in transmission rate of infection of directly exposed animals in their data and the second item also appears to be rather weak.

Response: We thank the reviewer for these suggestions. For the first comment regarding the transmission, we repeated the transmission studies with the SARS-CoV-2 Omicron BA.5 variant based on the reviewer's comment. As the new data illustrate, hamsters vaccinated with NanoSTING-SN showed 100% protection in preventing transmission to naïve animals.

Added figure (Pg. 30): Figure 5

Added text (Pg. 9-10, Line 298-311): “The Omicron variants are transmitted very efficiently, and we next wanted to directly investigate if immunization with NanoSTING-SN can prevent the transmission of this highly infectious VOC.....
.....These results demonstrate that NanoSTING-SN is highly effective at preventing the transmission of the Omicron VOC, which has implications for controlling the outbreak of respiratory pathogens”.

I don't see a reduction in transmission rate of infection of directly exposed animals in their data and the second item also appears to be rather weak.

Response: For the second comment regarding the magnitude of protection against SARS-CoV, we have repeated the experiments with two doses of NanoSTING-SN. As our experiments illustrate, vaccinated animals showed completely normal weights that were no different from unchallenged animals at all indicated time points. Unlike the unvaccinated animals, 100% of the vaccinated animals survived (Figures 6 and S14).

Updated figure (Pg. 32): Figure 6

Added figure (Pg. 49): Figure S14

1. The authors listed anti-RBD ELISA titers as a surrogate for neutralization without reference or validation of this position.

Response: We have added the references to the manuscript (PMIDs: 33399823 and 34971839).

Modified line number: Line 120

2. SARS-CoV-2 isolates are not referenced properly in terms of source (BEI number, etc).

Response: We have consistently referenced the VOCs and have included the BEI number in the methods section.

3. It looks like the authors recycled data from their PBS mock control and baseline control hamsters in Figures 1, 3, and 4 but did not clearly mention this in the text. If the intention is to compare immunogenicity and efficacy between NanoSting-S and NanoSting-SN, why not plot the two groups of data side by side? As a matter of fact, based on Fig. 1N and Fig. 3M, the NanoSting-S appears to reduce lung pathogen to a higher extent than NanoSting-SN.

Response: The reviewer is correct in that we used sera from the same PBS-treated animals, but the ELISA was repeated with each set of samples. We have added this information to the methods section.

Added text (Pg.17, Line: 545-546): “We used sera from the same PBS-treated animals, but the ELISA was repeated with each set of samples”.

As opposed to immunogenicity, the key comparison is the efficacy of protection against viral challenge in the hamster model, and these are already plotted together. As requested, the figure shown below compares the pathology scores of NanoSTING-S and NanoSTING-SN in one graph.

Figure: Comparisons of the pathology and infectious viral loads in hamsters vaccinated with NanoSTING-S, NanoSTING-N, NanoSTING-SN or placebo.

- (A) Pathology scores of the lung showing histopathological changes at day 6.
 (B) Viral titers measured by plaque assay in lungs post day 2 and day 6 of infection.
 (C) Viral titers measured by plaque assay nasal tissues post day 2 and day 6 of infection.

Vertical bars show mean values with error bar representing SEM. Each dot represents an individual hamster. Asterisks indicate significance between the groups. Significance testing was performed using a Mann-Whitney test comparing pairs of groups. The dotted line in graph B and C indicates LOD.

**** $p < 0.0001$; *** $p < 0.001$; ** $p < 0.01$; * $p < 0.05$; ns: not significant.

Integrated interpretation of pathology and viral load. The pathology scoring of the lung tissue was based on the inflammatory response in the bronchioles, alveoli and around the vessels. The high pathology score or reduction in pathology score compared to placebo was interpreted as a response to the two proteins, spike and nucleocapsid protein together (for NanoSTING-SN) compared to the inflammatory response to a single protein, spike protein (NanoSTING-S) alone (panel A). In the lung, both vaccines lead to a complete elimination of viral loads by day 6 (panel B). In the nasal turbinates, NanoSTING-SN immunization led to significantly lower infectious virus at day 2 (~4000-fold reduction vs placebo, and 10-fold reduction vs NanoSTING-S); and by day 6 the virus was undetectable in the NanoSTING-SN vaccine group (~8000-fold reduction vs placebo, and 40-fold reduction vs NanoSTING-S) [panel C]. Considering the complete elimination

of virus in the NanoSTING-SN vaccinated animals, the higher pathology score attributed to response to two proteins was interpreted in combination with the viral load reduction, thereby NanoSTING-SN was chosen for further development.

Antibody-dependent enhancement (ADE) and antibody-enhanced disease (AED). ADE/AED in viral infections are characterized by two immunological features (*e.g.* PMID: 32908214): (1) Th2 biased response, and (2) increase in the viral load. As our data clearly demonstrates (**Figure 3F-G**), we do not observe a Th2 response. As illustrated with our viral load data (**Figure 3J-K**), we observe an elimination in virus upon vaccination with NanoSTING-SN. As noted above, the higher pathology score attributed to response to two proteins was interpreted in combination complete elimination of the virus and NanoSTING-SN was chosen for further development.

Added text (Pg.8, Line: 238-242): “Although the resolution of the inflammatory responses was different comparing NanoSTING-SN (multiple antigens) vs NanoSTING-S (single antigen) [**Figure 3M and 1N**], viral titers in both the lung and nasal compartments were eliminated upon vaccination with NanoSTING-SN. In aggregate, these results illustrate that NanoSTING-SN can provide sterilizing immunity.”

Modified method section (Pg. 19, Line: 635-640): We used integrated scoring rubric for evaluating the pathology score⁷². The scoring method in the reference was modified from a 0-3 to a 0-4 score with 1= 1-25%, 2=26-50 %; 3=51-75; 4=76-100%, so with the three criteria mentioned in the reference will yield a score for an animal ranging from 0-12. This scoring also takes into account the degeneration/necrosis of the bronchial epithelium/alveolar epithelium. A board-certified pathologist evaluated the sections.

4. I am not clear how Fig. 2 adds anything.

Response: We have copy pasted the elegant summary from reviewer 3, “It is very pleasing to see what kind of central role the mathematical modelling part holds in this manuscript. This is encouraging for interdisciplinary work and shows how experimental work can benefit from mathematical models.”

5. Figure 4C: Y-axis is log₂ instead of log₁₀ like the other figures, is there a particular reason?

Response: We chose the scale to better illustrate the spread of the data points. We emphasize that we have made no comparisons on the magnitude of IgG titers between the serum and BALF.

6. Figure 5B: The transmission observed by the authors is 75% at d2 and 100% at d5 for both vaccinated and mock-vaccinated hamsters, and although a case can be made for reduced disease in contact hamsters the amount of “shed” virus in the nasal washes does not seem to be significantly reduced in the vaccinated hamsters. “Ending transmission” is clearly an overstatement.

Response: We thank the reviewer for this suggestion. We repeated the transmission studies with the SARS-CoV-2 Omicron BA.5 variant based on the reviewer's comment. As the new data

illustrate, hamsters vaccinated with NanoSTING-SN showed 100% protection in preventing transmission to naïve animals.

Added figure (Pg. 30): Figure 5

Added text (Pg. 9-10, Line 298-311): “The Omicron variants are transmitted very efficiently, and we next wanted to directly investigate if immunization with NanoSTING-SN can prevent the transmission of this highly infectious VOC.....
.....These results demonstrate that NanoSTING-SN is highly effective at preventing the transmission of the Omicron VOC, which has implications for controlling the outbreak of respiratory pathogens”.

7. The authors may perform a comparative study between intramuscular and intranasal vaccination using their vaccine candidates to demonstrate the advantage of the latter.

Response: We thank the reviewer for this suggestion. NanoSTING has been designed as a mucosal adjuvant, and hence, the formulation is not optimized for systemic delivery. Furthermore, as outlined in the discussion section, there are plenty of vaccines/vaccine candidates that have tested efficacy through the IM route. Hence, we did not duplicate these studies.

Reviewer #3 (Remarks to the Author):

I would like to congratulate you to this very nicely written manuscript. It is very pleasing to see what kind of central role the mathematical modelling part holds in this manuscript. This is encouraging for interdisciplinary work and shows how experimental work can benefit from mathematical models.

A possible vaccine strategy against S and N proteins of SARS-COV2 was mathematically tested and revealed that a synergetic effect would result in the best outcome - reducing viral titres of SARS-COV2. An intranasal S+N-vaccine has been designed and then tested in several animal models. The mathematical “prediction” could be approved, and the vaccine efficiency has been further tested with positive outcome regarding transmission and cross-reactivity.

Response: We thank the reviewer for their many positive comments and highlighting the complementary role of mathematical modeling in designing vaccines.

Major comments:

- Is there enough experimental data available, so that the model (derived on human data) can be transferred to the related animal model? It would be nice to see if an animal derived model could capture the experimental data. If this is not possible, please discuss.

Response: We agree with the reviewer that a mathematical model of SARS-CoV-2 in hamsters would delineate viral dynamics in further detail. There are published models on the viral dynamics in hamsters (PMID: 35913988, 36074824). While we could model the SARS-CoV-2 dynamics in unimmunized animals by using parameters published in above studies, we could not fit these models for vaccinated animals. Our viral load data was obtained from nasal tissue homogenates at

euthanasia and hence is not amenable to repeat sampling. We have measured viral titers at only two data points. Thus, the parameter space is very large for best fit, hence the confidence interval is very large. We envision that in the future, we will perform repeat sampling with nasal washes from the same animals and this can allow us to construct a vaccine-induced immunity model.

- The mathematical model suggests (according to Figure S5C) a lower viral titer for S+N vaccination, but the persistence of the viral load seems to be prolonged. Can that also be observed experimentally? Can you explain the basis of this result?

Response: We would like to thank the reviewer for this insightful comment. We did not see prolonged presence of viral titers in S+N vaccine model in hamsters, in fact viral particles could not be detected in S+N vaccination at day 6. The prediction of prolonged presence of viral particles might be an artefact of the model assumption that the elimination rate of infected cells is constant during infection. As the infection progresses, innate immune cells like macrophages and NK cells are activated, which eliminate infected cells at a much higher rate.

- Our model does not consider the effect of interferons on the rate constant of viral production by infected cells. As interferons are known to suppress viral production rate, the rate constant π for viral production should reduce as the infection progresses.
- Our model in Figure S5C predicts that the number of viral particles is between 1-100 which is below the experimental detection limit and hence cannot be confirmed experimentally.

We have modified our methods section to describe the limitations of the model.

Added text (Pg. 19, lines 646-651): “There are some limitations to our model. This model does not and hence cannot be confirmed experimentally.”

Minor comments:

- Line 128: Why was not also tested for beta IgA?

Response: During the period when these studies were conducted, the prevailing and most virulent strain was the Delta variant of concern (VOC), and we had planned challenge studies with Delta VOC because it is heterologous to the immunogen and provides us with the opportunity to test cross-protection.

- Line 129/130: It is not intuitive how IFN γ and IL4 translate to Th1/TC1 responses

Response: Thank you for your feedback. IFN- γ is associated with Th1/Tc1 responses, while IL-4 is linked to Th2 responses. This relationship is well-established in literature (PMID: 2523712, 8893001)

- Line 174: The N protein is chosen for what? Context is not completely clear; it might help to switch this sentence with the following.

Response: We have modified the text in the manuscript file.

Modified text (Pg.6, Line 173-177): “The results from the NanoSTING-S experiments demonstrated that the immune responses protect against disease in the lung but are insufficient to eliminate viral infection/replication in the nasal passage as a surrogate for transmission. To further bolster the protection against virus, we explored other antigens. We specifically chose the N protein because it is abundantly expressed soluble and immunogenic protein”.

- Lines 177-179: This sentence is very confusing and requires further explanation. As far as I am aware there is no parameter fitting performed in this manuscript – rather, you explored the parameter space manually. Of which infected patients are we talking here otherwise?

Response: We thank the author for pointing this out. We agree that we have not fitted any parameters in this work, and we have used the parameters for human viral kinetic data already published (PMID: 34857628). We have explored the parameter space to predict the effects of different modes of vaccination.

Modified text (Pg. 6-7, Line 179-181): “We explored the parameter space of an already published model describing viral kinetics in the nasal passage obtained by fitting longitudinal viral titers from infected patients (**Figure 2A**)”.

- Lines 179-187: It is not clear from the text if this part is based on S-protein only? I initially thought, it's the response against S+N.

Response: We have modified the text and the figure to better communicate immunity against each protein.

Modified figure (Pg.23): Figure 2

Modified text (Pg. 6-7, Line 179-186): “We explored the parameter space of an already established model describing viral kinetics in the nasal passage obtained by fitting longitudinal viral titers from infected patients (**Figure 2A**)⁴⁵.....The model revealed a reduction in viral load between 35 % to 90 % when the S-vaccine efficacy in the nasal compartment varied from 40 to 80 % (**Figure 2B**).

- Line 683: N+S (otherwise confusion with not significant).

Response: We have made the necessary changes in the manuscript file.

Modified line number: Line 641

- Figure1: SEM – standard error on the mean? Please be specific and introduce abbreviation.

Response: We have made the necessary changes to the manuscript file.

- *Figure2: Is it possible to separate the figure in 4 sections (I: no vaccination, II:S only, III: N only, IV: S+N)? It would be easier to refer to in the main text as well as more understandable for the reader.*

Response: We have updated the figure and added the appropriate titles to each plot to clearly convey the immunogen and immunity.

Updated figure (Pg. 24): Figure 2

- *Figure2A: The schematic illustration requires a proper description in the legend.*

Response: We thank the author for this suggestion. We have made the following changes in the figure legend.

Modified text (Pg. 24 Line: 730-740): “Schematic and governing equations describing viral dynamics without vaccination, with spike protein immunization, or nucleocapsid protein immunization (IFNAR: interferon- α/β receptor, IFN1: type-I interferons, ISG: interferon-stimulated gene.....Upon immunization with N protein, the rate constant of elimination of infected cells is increased by ω due to killing of infected cells by T cells”.

- *Figure2: AUC – area under the curve*

Response: We have modified the figure legend and explained AUC.

Modified text (Pg.24, Line: 741-742): “Percent reduction in viral area under the curve (AUC) with increasing de-novo blocking efficiency (antibodies against the spike protein)”.

- *Figure2C: Does omega vary between 0 and 1 or 0 and 10?*

Response: ω represents a bulk parameter representing the killing rate constant of infected cells by T cells, in the figure 2C, it varies from 0 to 10.

- *Figure2D: Why is now no longer reduction in viral AUC used as a measure?*

Response: When the efficiency of antibody mediated protection is high ($\gamma = 0.8-1$), the percent reduction in viral AUC varies between (95%-100%). When the efficiency of antibody mediated protection is moderate ($\gamma = 0.4-0.7$), the percent viral reduction drops (34-70 % reduction in viral AUC). To illustrate the concept that even when antibody responses are high ($\gamma = 0.8$), T cell mediated killing of infected cells could reduce viral load further and hinder transmission of viral particles. But as shown in the figure below, visualizing reduction in viral AUC for high ($\gamma = 0.8$) with varying ω is difficult because the variation in AUC is very low (94-100%) compared to variation in AUC in the whole heatmap (34-100%).

Thus, to effectively communicate the importance of T-cell responses when the antibody responses are high, we chose to show peak viral load. Furthermore, peak viral loads are a better measure for viral transmission as compared to percent reduction in viral AUC (PMID: 34857628).

Figure: Variation in reduction of viral area under the curve (AUC) with varying antibody blocking efficiency (γ) and T cell mediated killing (ω).

- *Figure3A: What is the reason that the protocol changed to euthanize the animals now on day 51 instead on day 28 (previous study)?*

Response: We extended the study for an additional three weeks since the immune responses do not significantly change during these relatively short time intervals.

- *Figure3B: Why was not tested against omicron IgG?*

Response: During the period when these studies were conducted, the prevailing and most virulent strain was the Delta variant of concern (VOC).

- *Figure4B: It is not clear against which strains was tested.*

Response: We tested it against the SARS-CoV2 Wuhan-Hu-1 strain [mentioned in the method section (Pg. 15, Line 479)].

- *FigureS5: The model output in panel A looks anomalous for gamma=0.8 (blue curve). Can you explain this? Does it eventually decline? Extension of the x-axis would be appropriate to see the full AUC.*

Response: We thank the reviewer for the comment. We agree that the curve for $\gamma = 0.8$ looks anomalous. This anomaly arises because just modifying infectivity term is not enough to capture the viral dynamics upon immunization with the spike protein. While the antibody responses act primarily by reducing the rate of viral infection, there are innate and adaptive responses during

later phases of infection which are not taken account in the model. In our model, it is assumed that the strength of antiviral response depends on the number of infected cells. As stated above, our model doesn't consider the innate immune component of the antiviral responses that become important during prolonged infections.

Extending the model out to a longer timepoint shows that the viral load decreases at later timepoints.

Figure: Evolution of viral titer dynamics for $\gamma = 0.8$ for extended time period (0-60 days)

To verify this observation in an independent dataset, we solved the model published in PMID: [33750978](https://pubmed.ncbi.nlm.nih.gov/33750978/) as they have studied the effect of blocking viral transmission via antibodies in their model for SARS-CoV-2. The viral titer data for blocking efficiency of 80% shows delayed and extended clearance similar to our model.

Figure: Evolution of viral dynamics with varying antibody blocking efficiency (γ) for the model published in PMID: [33750978](https://pubmed.ncbi.nlm.nih.gov/33750978/). Even in other models published in literature, viral titer clearance is extended and delayed.

- FigureS5: A more detailed legend is required.

Response: We thank the reviewer for this suggestion. We have written a more detailed legend for figure S5 as shown below:

Modified figure legend (Pg.40, Line:919-927): “Figure S5: Evolution of viral dynamics with

- (A) S immunization (assuming only de-novo blocking of viral entry). Increasing the de-novo blocking efficiency of viral entry into target cells decreased the rate of viral infection. Blocking efficiency of 80% ($\gamma = 0.8$) significantly reduced the viral titer growth rate.
- (B) N immunization (assuming only cytotoxic T cell killing of infected cells). T cell responses alone do not reduce the viral load significantly.
- (C) Immunization with both N and S combined. Physiological rates of T cell responses ($\omega = 0-0.6 \text{ day}^{-1}$) with de-novo blocking of viral entry with 80% efficiency ($\gamma = 0.8$) act synergistically to reduce viral replication (related to Figure 2)”.

- FigureS5: Replace (D) by (C) in legend.

Response: We thank the reviewer for pointing out this error. We have corrected the legend.

Reviewers' Comments:

Reviewer #1:

Remarks to the Author:

Authors have addressed most of my comments. The quality of this paper was significantly improved. I have no other comments.

Reviewer #2:

Remarks to the Author:

There are multiple differences between contact transmission studies which warrant both being included in the final manuscript. Initially, you tested transmission of B.1.1.529 Omicron and tested contact and index hamsters for viral titers in nasal tissue. In the new experiment you use an updated Omicron isolate (BA.5) but measure only lung viral titers. Both studies should be included in the final manuscript, with lung AND nasal tissue titers presented for both experiments. Overall, the cumulative evidence presented does not justify the claim that your vaccine prevents transmission.

You use the anti-RBD ELISA (the one claimed to correlate with neutralization) sparingly: only against alpha and/or beta, only in sera alone for mice (figure 1, 3), not for other variants as you do for your full-length spike ELISA. In your methods, you don't mention the RBD ELISA, only the full-length spike and nucleocapsid ELISA. One of their reasons stated for using ELISA opposed to neutralization assays is that you can test against more variants, however, the RBS ELISA is not used enough to justify that claim. The RBD ELISA is not mentioned at all for the NHP studies (Figure 6) or in S4/S14. More evidence is needed to claim that your ELISA results represent functional antibody titers in vaccinated animals.

Additionally, PMID 33399823 has the following caveat for correlating ELISA to VN: "None of commercial assays have sufficient performance to detect a neutralizing titer of 80 (AUC<0.76)" which is a weakness in your study regarding the detection of low titers. Your methods should include the commercial assay used for anti-RBD ELISA, which should be similar to the referenced studies you use for justification.

The authors acknowledged that they used sera from the same PBS-treated animals in Figs 1, 3 & 4. How were animals in all these studies age matched then?

BEI requests detailed attributions such as the example below for NR-55612 (your Delta variant isolate):

Acknowledgment for publications should read "The following reagent was obtained through BEI Resources, NIAID, NIH: SARS-Related Coronavirus 2, Isolate hCoV-19/USA/PHC658/2021 (Lineage B.1.617.2; Delta Variant), NR-55611, contributed by Dr. Richard Webby and Dr. Anami Patel."

You use an end-point titration assay (CCID50) but call it a plaque assay. Figure legends need to be corrected for clarity.

You cannot have sterilizing immunity if there is detectable virus in vaccinated animals post-challenge. Sterilizing immunity is sufficient to prevent viral replication in the host after infection.

Reviewer #3:

Remarks to the Author:

The authors have addressed all my concerns. However, I still have my concerns about the presentation

of the mathematical model in Figure 2. Figure 2 is stuffed with information, and the reader requires some more structure to be guided through (see also Reviewer #2). Particularly, panel A might need some subheadings. Also, the panels B, C and D could benefit from some more space between the panels to clearly separate the graphs from each other. Try to link them back to the models presented in panel A.

The legend has improved a lot, even though the parameters ρ , k and σ are still not introduced. A refer to the supplementary method might be helpful here.

The authors have added limitation in the method sections, which is fine. However, the second point (Second, our model in Figure S5C predicts that the number of viral particles is between 1-100 which is below the experimental detection limit and hence cannot be confirmed experimentally.) could also easily stated in the figure legend itself; or can just be made visible by adding a threshold line in the figure.

Reviewer #1 (Remarks to the Author):

Authors have addressed most of my comments. The quality of this paper was significantly improved. I have no other comments.

Response: Thanks for the feedback. We appreciate your valuable input.

Reviewer #2 (Remarks to the Author):

There are multiple differences between contact transmission studies which warrant both being included in the final manuscript. Initially, you tested transmission of B.1.1.529 Omicron and tested contact and index hamsters for viral titers in nasal tissue. In the new experiment you use an updated Omicron isolate (BA.5) but measure only lung viral titers. Both studies should be included in the final manuscript, with lung AND nasal tissue titers presented for both experiments. Overall, the cumulative evidence presented does not justify the claim that your vaccine prevents transmission.

Response: As requested by the reviewer, we have added viral titers for both the nasal turbinates and the lung in the SARS-CoV2 Omicron VOC (BA.5) transmission study.

Updated Figure: Figure 5, Pg. 32

Added text (Pg. 10, Line: 311-314): Vaccination protected the index hamsters from the virus with only 2/8 hamsters showing a low amount of detectable virus in the lung (**Figure 5B**) and nasal tissue (**Figure 5D**). Vaccination completely blocked transmission in the lungs (**Figure 5C**) and nasal tissue (**Figure 5E**), and none of the contact hamsters showed detectable virus.

The editor during our first discussion had suggested to replace the single dose vaccine studies with the dual dose vaccine studies to be consistent with the rest of the manuscript. We have accordingly performed these studies with the BA.5 variant. Since the original study with the SARS-CoV2 B.1.1.529 variant was based on a single dose vaccine and all the challenge studies in the entire manuscript are based on two doses of vaccines, we respectfully disagree that we should have to include it in this updated manuscript. We thank both the reviewer and the editor for the suggestion to improve the manuscript and keep it consistent.

You use the anti-RBD ELISA (the one claimed to correlate with neutralization) sparingly: only against alpha and/or beta, only in sera alone for mice (figure 1, 3), not for other variants as you do for your full-length spike ELISA. In your methods, you don't mention the RBD ELISA, only the full-length spike and nucleocapsid ELISA. One of their reasons stated for using ELISA opposed to neutralization assays is that you can test against more variants, however, the RBS ELISA is not used enough to justify that claim. The RBD ELISA is not mentioned at all for the NHP studies (Figure 6) or in S4/S14. More evidence is needed to claim that your ELISA results represent functional antibody titers in vaccinated animals.

Response: Unfortunately, it appears that the reviewer may have overlooked the data presented in Figures 7D & 7F. In both these figures, we have provided results from anti-RBD IgG and IgA ELISA assays conducted on non-human primates (NHPs). Importantly, these assays were performed using two separate SARS-CoV2 Omicron VOCs (XBB1.5 and BA.1), that were the predominant variants at the time of our study. We have made the necessary changes in the method section for ELISA.

Added text for mouse RBD-ELISA (Pg.17, Line 556-573): We evaluated RBD-specific serum IgG and IgA in mice using indirect ELISA.....Finally, we recorded optical density (OD) values using Cytation 7 (Biotek Instruments Inc).

Added text for NHP RBD-ELISA (Pg.21, Line 708-726): We evaluated RBD-specific serum IgG and IgA in NHPs using indirect ELISAs.....We recorded optical density (OD) values using Cytation 7 (Biotek Instruments Inc).

Additionally, PMID 33399823 has the following caveat for correlating ELISA to VN: “None of commercial assays have sufficient performance to detect a neutralizing titer of 80 (AUC<0.76)” which is a weakness in your study regarding the detection of low titers. Your methods should include the commercial assay used for anti-RBD ELISA, which should be similar to the referenced studies you use for justification.

Response: We recognize the broader point that RBD ELISAs are only a surrogate for pseudovirus neutralization assays. As stated above, the use of RBD ELISA allowed us to test many different variants across multiple species.

We would also like to draw the reviewer’s attention to a separate study that describes a surrogate virus neutralization test (sVNT) for detecting neutralizing antibodies (NAbs) without live virus or cells. The test mimics the conventional virus neutralization test (VNT) by using the interaction between RBD and ACE2. In this study, the authors also correlate RBD-binding antibodies measured by ELISA (indirect RBD ELISA) with neutralizing antibody levels from the sVNT, supporting the use of RBD-binding antibodies as a surrogate marker for neutralization. The graph below shows the correlation of SARS-CoV-2 sVNT and indirect RBD ELISA (PMID: 32704169). We have added this reference to our manuscript.

“Correlation of SARS-CoV-2 sVNT and indirect RBD ELISA: Pearson’s correlation coefficient and linear regression analysis was performed using end-point titer of SARS-CoV-2 sVNT and ELISA using the same 60-serum panel as that in Fig. 4. Dashed line indicates the standard deviation from the linear regression analysis. Statistical significance was determined using the two-tailed test.”

The authors acknowledged that they used sera from the same PBS-treated animals in Figs 1, 3 &4. How were animals in all these studies age matched then?

Response: To avoid any confusion, we have retested sera from the control animals that were administered PBS during the same time as the respective vaccines. We have confirmed the endpoint-titers for each set of control mice and updated the method section accordingly.

Updated text (Pg.17, Line:552-554): “We conducted three individual studies (NanoSTING-S, NanoSTING-N and NanoSTING-SN), each with its own distinct control sets, and collected sera from the control animals upon completion of each study”.

BEI requests detailed attributions such as the example below for NR-55612 (your Delta variant isolate):

Acknowledgment for publications should read “The following reagent was obtained through BEI Resources, NIAID, NIH: SARS-Related Coronavirus 2, Isolate hCoV-19/USA/PHC658/2021 (Lineage B.1.617.2; Delta Variant), NR-55611, contributed by Dr. Richard Webby and Dr. Anami Patel.”

Response: We thank the reviewer for this reminder and have added the detailed attributions for all the isolates used in the study to the acknowledgements section.

Added text (Pg. 21, Line: 735-746): NR-55612: SARS-Related Coronavirus 2 Isolate hCoV-19/USA/PHC658/2021 (Lineage B.1.617.2; Delta Variant).....This virus was adapted through 25 serial passages in the lungs of mice and plaque-purified for use in mouse infections⁷³.

You use an end-point titration assay (CCID50) but call it a plaque assay. Figure legends need to be corrected for clarity.

Response: The figure legends have been corrected in the updated manuscript file.

You cannot have sterilizing immunity if there is detectable virus in vaccinated animals post-challenge. Sterilizing immunity is sufficient to prevent viral replication in the host after infection.

Response: Our data from Figure 3K & 3J demonstrate that NanoSTING-SN effectively eliminates viral replication in both lung and nasal compartments post-challenge by day 6. In any intranasal challenge experiment wherein a high dose of virus is administered (3×10^5 CCID in our case), it is unlikely that all the virus from the inoculum has been eliminated at early timepoints like day 2. It is for this reason that the day 6 readout is essential to map true viral replication and distinguish it from the inoculum. As the reviewer suggested, since there is detectable virus in 2 animals upon challenge with the SARS-CoV2 Omicron VOC, we have ensured that there are no claims of sterilizing immunity with the SARS-CoV2 Omicron VOC.

Reviewer #3 (Remarks to the Author):

The authors have addressed all my concerns. However, I still have my concerns about the presentation of the mathematical model in Figure 2. Figure 2 is stuffed with information, and the reader requires some more structure to be guided through (see also Reviewer #2). Particularly, panel A might need some subheadings. Also, the panels B, C and D could benefit from some more space between the panels to clearly separate the graphs from each other. Try to link them back to the models presented in panel A.

The legend has improved a lot, even though the parameters rho, k and sigma are still not introduced. A refer to the supplementary method might be helpful here.

The authors have added limitation in the method sections, which is fine. However, the second point (Second, our model in Figure S5C predicts that the number of viral particles is between 1-100 which is below the experimental detection limit and hence cannot be confirmed experimentally.) could also easily stated in the figure legend itself; or can just be made visible by adding a threshold line in the figure.

Response: We thank the reviewer for these comments. We have reorganized Figure 2 (Line as suggested by the reviewer. We have updated the figure legend and introduced the remaining parameters. We have further redirected the readers to supplementary data.

We have introduced a dashed line in the figure to indicate experimental limit of detection and described the limit in figure legend.

Updated figures (Pg. 25 & 41): Figures 2 & S5

Updated figure legend (Figure 2; Line 795-815): Schematic and governing equations describing viral dynamics without vaccination, with spike protein immunization, or nucleocapsid protein immunization (IFNAR: interferon- α/β receptor, IFN1: type-I interferons, ISG: interferon-stimulated gene).....

.....The heatmap shows the effectiveness of combined effect of de-novo blocking (S response) and T cell-mediated killing (N response). The red box indicates the synergistic effect of N and S response in achieving sterilizing immunity.

Updated figure legend (Figure S4; Pg. 40, Line 992-1000): S immunization (assuming only de-novo blocking of viral entry). Increasing the de-novo blocking efficiency of viral entry decreased the rate of viral infection..... Physiological rates of T cell responses ($\omega = 0-0.6 \text{ day}^{-1}$) with de-novo blocking of viral entry with 80% efficiency ($\gamma = 0.8$) act synergistically to reduce viral replication (related to Figure 2). Dashed red line indicates experimental limit of detection of viral titers.

Reviewers' Comments:

Reviewer #2:

Remarks to the Author:

I appreciate the efforts of the authors to improve the quality of the manuscript. Pasted below please find my comments:

1. The manuscript is still filled with language that needs to be revised. A few examples are:

Line 302, the authors state "NanoSTING-SN abolishes transmission of SARS-CoV-2 Omicron VOC". Yet the data reported in Figure S13 essentially contradicts with this title. I'd suggest removing phrases like "abolish or eliminate transmission" from this manuscript. It has been recognized from the early pandemic that natural infection acquired immunity could temporarily restrict virus transmission, so did the mRNA vaccines. However, we soon realize that such an effect has a very narrow time window. Performing a transmission study 2- or 3-weeks post-vaccination when the circulating neutralizing antibody titer is at its peak could easily derive such conclusions, but we now know they are misleading. If the authors insist on the use of such phrases, please qualify the statement by citing time frame of the study.

Line 352 and the caption of Figure 7 state "NanoSTING-SN confers long term humoral immunity in Rhesus macaques". Keep in mind detectable antibodies after 6 months of vaccination is not long-term. It is probably the minimum that one should expect from a vaccine candidate. I did not comment on Figure 7, but since the authors mentioned Figure 7 in their rebuttal, please include limit of detection or quantification in figures where you are reporting results from ELISA assays. Please do not lump sum anti-N antibody titers and anti-S or anti-RBD antibody titers in the same panel with the same "control bar" because each ELISA assay (against a different target) would have a different limit of quantification (cutoff).

Figure 1. NanoSTING-S reduced infectious viral loads in the lung by 300-fold by day 2 compared to sham-vaccinated animals, and by day 6, infectious virus was undetectable in all animals. The fold of reduction is not superior to many reported (for example, intranasal vaccination with mRNA vaccine reduced to greater extents. *Sci Adv.* 2023 Sep 22; 9(38): eadh1655). In a different publication (*Nat Commun* 13, 6644 (2022). <https://doi.org/10.1038/s41467-022-34439-7>) on this journal, the authors demonstrated vaccinated hamsters did not transmit Delta variant to non-vaccinated cage mates when challenged three weeks after vaccination.

The authors interestingly drew illustrations featuring trimeric Spike protein embedded to the surface of liposomal particles (Fig. S2). Have the authors performed CryoEM or other experiments to demonstrate that the spike protein decorates the liposome surface as how it was drawn?

REVIEWERS' COMMENTS

Reviewer #2 (Remarks to the Author):

I appreciate the efforts of the authors to improve the quality of the manuscript. Pasted below please find my comments:

1. The manuscript is still filled with language that needs to be revised. A few examples are:

Line 302, the authors state "NanoSTING-SN abolishes transmission of SARS-CoV-2 Omicron VOC". Yet the data reported in Figure S13 essentially contradicts with this title. I'd suggest removing phrases like "abolish or eliminate transmission" from this manuscript. It has been recognized from the early pandemic that natural infection acquired immunity could temporarily restrict virus transmission, so did the mRNA vaccines. However, we soon realize that such an effect has a very narrow time window. Performing a transmission study 2- or 3-weeks post-vaccination when the circulating neutralizing antibody titer is at its peak could easily derive such conclusions, but we now know they are misleading. If the authors insist on the use of such phrases, please qualify the statement by citing time frame of the study.

Response: We have corrected the sub-header to read "Two doses of NanoSTING-SN abolishes transmission of SARS-CoV-2 Omicron VOC". As we had referenced in our Discussion section, prior work with mRNA vaccines (testing both single and dual antigens), showed that dual antigen (S and N) mRNA vaccines did not eliminate the viral load in the nasal compartment when challenged with either the Delta or Omicron VOC. The challenge was conducted two weeks after the booster dose. (Figures 3H & 4H, <https://www.science.org/doi/full/10.1126/scitranslmed.abq1945>).

As suggested by the reviewer, we have added the following sentence to our Results section, "These results suggest that at least early after immunization, two doses of NanoSTING-SN can prevent transmission of highly transmissible variants and long-term studies are warranted to quantify the duration of this protection."

Corrected text: Pg. 18, Line 337

Added text: Pg.19, Line 352-354

Line 352 and the caption of Figure 7 state "NanoSTING-SN confers long term humoral immunity in Rhesus macaques". Keep in mind detectable antibodies after 6 months of vaccination is not long-term. It is probably the minimum that one should expect from a vaccine candidate. I did not comment on Figure 7, but since the authors mentioned Figure 7 in their rebuttal, please include limit of detection or quantification in figures where you are reporting results from ELISA assays. Please do not lump sum anti-N antibody titers and anti-S or anti-RBD antibody titers in the same panel with the same "control bar" because each ELISA assay (against a different target) would have a different limit of quantification (cutoff).

Response: As suggested by the reviewer, we have corrected the sub-header to read "NanoSTING-SN confers durable humoral immunity in Rhesus macaques". We have split the ELISAs against each protein into separate subpanels in Figure 8. Additionally, for ELISA, we have added the following sentence to our method section, "The threshold for positivity was set at an optical density (450 nm) value of 0.05 or two times the negative control (PBS), whichever was

higher. Endpoint titers were defined as the lowest dilution that was higher than the threshold for positivity”.

Corrected text: Pg. 21, Line 394

Added text: Pg. 42-43, Line: 814-817

Corrected figure: Figure 8

Figure 1. NanoSTING-S reduced infectious viral loads in the lung by 300-fold by day 2 compared to sham-vaccinated animals, and by day 6, infectious virus was undetectable in all animals. The fold of reduction is not superior to many reported (for example, intranasal vaccination with mRNA vaccine reduced to greater extents. Sci Adv. 2023 Sep 22; 9(38): eadh1655). In a different publication (Nat Commun 13, 6644 (2022). <https://doi.org/10.1038/s41467-022-34439-7>) on this journal, the authors demonstrated vaccinated hamsters did not transmit Delta variant to non-vaccinated cage mates when challenged three weeks after vaccination.

Response: We agree with the reviewer that several other vaccines/vaccine candidates including the mRNA intranasal vaccine paper (Sci Adv. 2023 Sep 22; 9(38): eadh1655) that show reduction in viral titers that are non-inferior to the results we report. This paper challenged the vaccinated with the ancestral SARS-CoV-2 Wuhan variant and quantified viral loads at day 3 (as opposed to day 2 in our study). Thus, it is not clear why the Sci Adv paper and our results are being compared head-to-head.

The second paper (Nat Commun 13, 6644 (2022). <https://doi.org/10.1038/s41467-022-34439-7>) studies transmission of the SARS-CoV-2 Delta VOC. Unfortunately, the paper does not report viral titers in the nasal compartment and only reports lung viral titers. Thus, it is very hard to conclude based on the results of the Nat Comm paper that transmission was completely blocked.

The authors interestingly drew illustrations featuring trimeric Spike protein embedded to the surface of liposomal particles (Fig. S2). Have the authors performed CryoEM or other experiments to demonstrate that the spike protein decorates the liposome surface as how it was drawn?

Response: We have previously published the data supporting the Spike protein on the surface of NanoSTING (PMID: [34462731](https://pubmed.ncbi.nlm.nih.gov/34462731/)).